# Hawks steer attacks using a guidance system tuned for close pursuit of erratically manoeuvring targets

Caroline H. Brighton[1] & Graham K. Taylor [1]

Aerial predators adopt a variety of different hunting styles, with divergent flight morphologies typically adapted either to high-speed interception or manoeuvring through clutter, but how are their sensorimotor systems tuned in relation to habitat structure and prey behavior? Falcons intercept prey at high-speed using the same proportional navigation guidance law as homing missiles. This classical guidance law works well in the open, but performs sub-optimally against highly-manoeuvrable targets, and may not produce a feasible path through the cluttered environments frequented by hawks and other raptors. Here we identify the guidance law of $n = 5$ Harris' Hawks *Parabuteo unicinctus* chasing erratically manoeuvring artificial targets. Harris' Hawks use a mixed guidance law, coupling low-gain proportional navigation with a low-gain proportional pursuit element. This guidance law promotes tail-chasing and is not thrown off by erratic manoeuvres, making it well suited to the hawks' natural hunting style, involving close pursuit of agile prey through clutter.

[1] Department of Zoology, University of Oxford, 11a Mansfield Road, Oxford OX1 3SZ, UK. Correspondence and requests for materials should be addressed to G.K.T. (email: graham.taylor@zoo.ox.ac.uk)

The relationship between form and function is nowhere clearer than in the morphology, physiology and behaviour of aerial predators—from the echolocation systems of bats, to the streamlined silhouette of a stooping raptor. Birds of prey adopt a range of hunting styles shaped by habitat structure, prey behaviour and flight performance[1]: hawks have broad, lightly loaded wings that enable them to turn quickly during short surprise attacks through clutter, whereas falcons have narrower and more heavily loaded wings suited to fast long-range attacks in open environments[2]. Flight performance is only a part of the story, however, because attack success also hinges on the feedback system used to guide flight after evasive prey[3]. The higher-level clades containing hawks and falcons diverged >60 mya, but as their common ancestor with other landbirds is also thought to have been an apex predator[4,5], it is reasonable to suppose that their raptorial guidance systems might share a common evolutionary origin. In level flight, Peregrine Falcons *Falco peregrinus* have recently been shown to use the same proportional navigation guidance law as missiles[6], which commands turning in proportion to the angular rate of the line-of-sight from attacker to target[7]. At high enough gain, this feedback system produces an attack trajectory called a parallel navigation (or constant absolute target direction, CATD) course, in which the geographic direction of the line-of-sight remains approximately constant, causing the attacker to intercept its target by heading it off. This is in contrast to the simpler geometry of a pure pursuit course, in which flight is always aimed directly at the target, causing the attacker to follow its target in a tail-chase. Hawks may be expected to use a different guidance law to Peregrine Falcons in level flight, because their attacks have been found to involve a mixture of different guidance behaviours, combining elements of parallel navigation with elements of pure pursuit[8], although this work did not attempt to identify the underlying feedback laws implementing these behaviours.

Here we identify the guidance law of Harris' Hawks *Parabuteo unicinctus* chasing erratically manoeuvring artificial targets. The hawks' measured attack trajectories approximate a delayed pure pursuit course, with flight directed at the location of the target after a short lag. Such pursuit kinematics could be produced by one of several different kinds of guidance law, including (i) proportional pursuit, in which turning is commanded in proportion to the deviation angle between the attacker's velocity vector and its line-of-sight to target, and (ii) proportional navigation, in which turning is commanded in proportion to the angular rate of the attacker's line-of-sight to target. In fact, we find that the dynamics are best modelled under a mixed guidance law, in which turning is commanded by feeding back a linear combination of the deviation angle and the line-of-sight rate with a small delay. Fitting the parameters of this mixed guidance law globally to all flights results in a closer prediction of the observed flight behaviour than does fitting the parameters of delayed proportional pursuit or delayed proportional navigation independently to each flight. We conclude by discussing how the structure and tuning of the hawks' mixed guidance law relates to their typical hunting style involving close pursuit of erratically manoeuvring targets, and consider how this compares with the proportional navigation guidance law used during similarly level chases by falcons specialised on hunting in open environments.

## Results

**Experimental procedure**. We used four high-speed cameras recording at 250 Hz to reconstruct the three-dimensional (3D) flight trajectories of $n = 5$ captive-bred Harris' Hawks *Parabuteo unicinctus* (Supplementary Table 1) during 50 flights against an erratically manoeuvring artificial target. The target comprised a food lure, which was towed at speed around a series of pulleys to create a sequence of zigzagging turns that we randomised on each trial to prevent the hawks from learning the course (see Methods). The speed of the lure was adjusted continuously by the experimenter (median lure speed: 7.0 m s$^{-1}$; interquartile range, IQR: $9.7 - 4.1$ m s$^{-1}$) to keep it ahead of the hawk until the moment of capture. The resulting motions were intended to mimic the erratic manoeuvres, or jinks, of a typical terrestrial prey item (e.g. a hare or jackrabbit, *Lepus* spp.). There is very limited information available on typical prey performance, and we did not attempt to model the behaviour of any particular prey item specifically, but as a point of reference, the lure's turning performance was broadly comparable to that of a European Hare *Lepus europaeus*, which has been recorded making a 60° evasive turn on a 7-m radius when fleeing a predator at 10 m s$^{-1}$[9].

**Hawk attack kinematics approximate a delayed pure pursuit**. The hawks took off as soon as the lure began moving, flap-gliding to reduce the range to their target. The birds banked to turn, whilst appearing to keep their eyes level and their gaze directed at the target (Fig. 1a–c; Supplementary Movies 1 and 2). The hawk rapidly extended one leg to capture the lure, typically placing the other foot on the ground (Fig. 1d). Because the recorded attack trajectories were always close to planar, we projected them onto the two-dimensional (2D) ground plane defined by the lure's trajectory prior to further analysis. The hawks' measured track angle $\gamma(t)$, defined as the bearing of their ground velocity vector in an inertial frame of reference (Fig. 2a), was usually similar (Fig. 2b) to their line-of-sight angle $\lambda(t)$, defined as the bearing of the position vector from hawk to lure (Fig. 2a). The linear association between $\gamma(t)$ and $\lambda(t)$ was not always very strong, however (Fig. 3a), so to account for expected sensorimotor delay, we also tried lagging the track angle by $\tau \geq 0$ (Fig. 3b). The resulting correlation between $\gamma(t)$ and $\lambda(t - \tau)$ for each flight usually peaked at $r > 0.8$, given a small delay with median value $\tilde{\tau} = 0.16$ s (IQR: $0.22 - 0.12$ s; here and elsewhere, tilde notation denotes the median value of a parameter). It follows that the trajectories approximated a delayed pure pursuit course, and not a parallel navigation course, as is obvious also by inspection of the line-of-sight plots in Fig. 2b. Although this simple correlation analysis does not provide an explicit model of the dynamics, the fitted delay is nevertheless comparable with the 0.13 s sensorimotor delay fitted in a steering controller used to model pigeons negotiating obstacles[10].

**Attack dynamics are best modelled by a mixed guidance law**. The statement that the hawks' attack trajectories approximated a delayed pure pursuit course is agnostic with respect to the feedback system that implements this. The simplest way to implement a pure pursuit course is to command turning in proportion to the deviation angle $\delta = \gamma - \lambda$ between the velocity vector and the line-of-sight to target (Fig. 2a), such that $\dot{\gamma}(t) = -K\delta(t - \tau)$ where $K > 0$ is a guidance constant[7]. This proportional pursuit (PP) guidance law drives $\delta$ to zero, so is a direct way of implementing pure pursuit. An indirect approach is to use a proportional navigation (PN) guidance law $\dot{\gamma}(t) = N\dot{\lambda}(t - \tau)$ with its guidance constant set at $N = 1$, which commands turning at a rate $\dot{\gamma}$ equal to the line-of-sight rate $\dot{\lambda}$, thereby holding $\delta = \gamma - \lambda$ unchanging[6]. This PN guidance law differs fundamentally from PP, in that it has no tendency to drive $\delta$ to a specific value, so whereas both will produce pure pursuit trajectories for $\delta(0) = 0$, they will produce different trajectories for other initial conditions. Yet another possibility is to use a mixed guidance law $\dot{\gamma}(t) = N\dot{\lambda}(t - \tau) - K\delta(t - \tau)$ combining PP and PN[7], which has already been tested in several other contexts in the missile

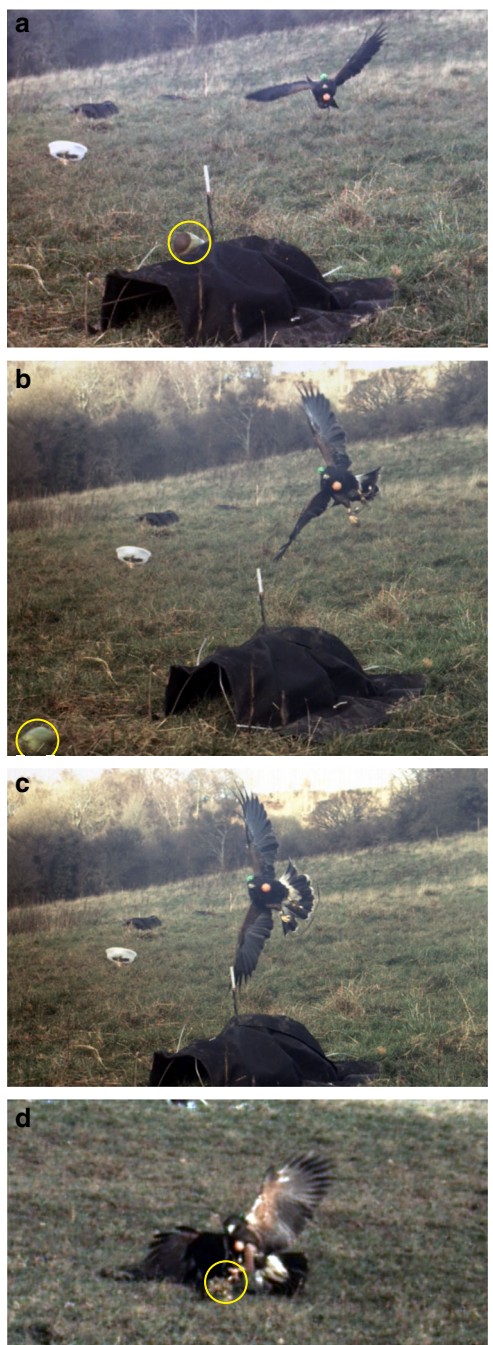

**Fig. 1** Sequence of video frames from a representative attack. Five captive Harris' Hawks chased a lure pulled at speed around a randomised zigzagging course of pulleys and tunnels. **a** Hawk initiating banked turn, with its inside wing pitched down and its outside wing pitched up; **b** hawk at a high bank angle 0.25 s later, with its head held such that the eyes are approximately level; **c** hawk fully banked another 0.05 s later, with its head still held with the eyes approximately level; **d** hawk captures lure with one foot. Lure is circled yellow if visible

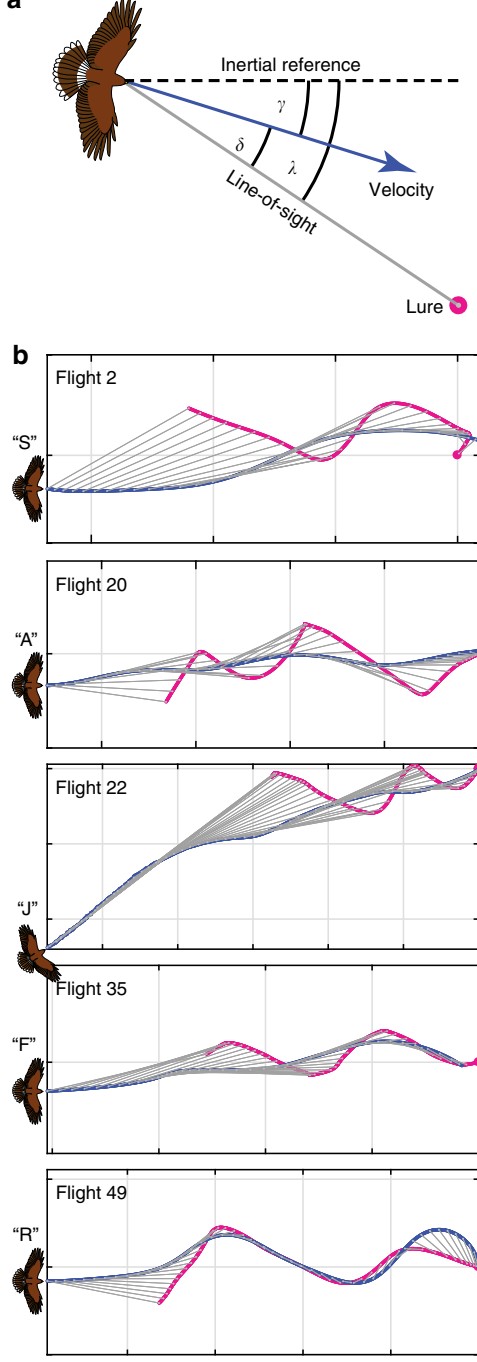

**Fig. 2** Geometry of a pursuit. **a** Definition sketch showing: $\lambda$, the line-of-sight angle measured between the line-of-sight (grey line) from hawk to target, and some arbitrary inertial reference direction (dashed line); $\gamma$, the track angle measured between the hawk's velocity vector (blue arrow) and the inertial reference direction (dashed line); $\delta$, the deviation angle measured between the line-of-sight from hawk to target (grey line), and the hawk's velocity vector (blue arrow). **b** 2D trajectory plots showing how the instantaneous line-of-sight (grey line) between the hawk (blue) and lure (magenta) varies through a flight for one randomly selected flight per bird. Note that although the line-of-sight is sometimes aligned with the direction of flight (as is expected under the geometry of pure pursuit), the direction of the line-of-sight itself is rarely held constant (as would have been expected under the geometry of parallel navigation). Grid spacing: 10 m

literature[11–13]. To test between these alternatives, we simulated all 50 flight trajectories under PP, PN and the mixed guidance law, given knowledge only of the initial track angle and position of the hawk, and the complete time history of both the lure's motion and the hawk's groundspeed (Fig. 4).

Because previous work on falcons had found variability in the guidance constant between flights[6], we began by fitting the time

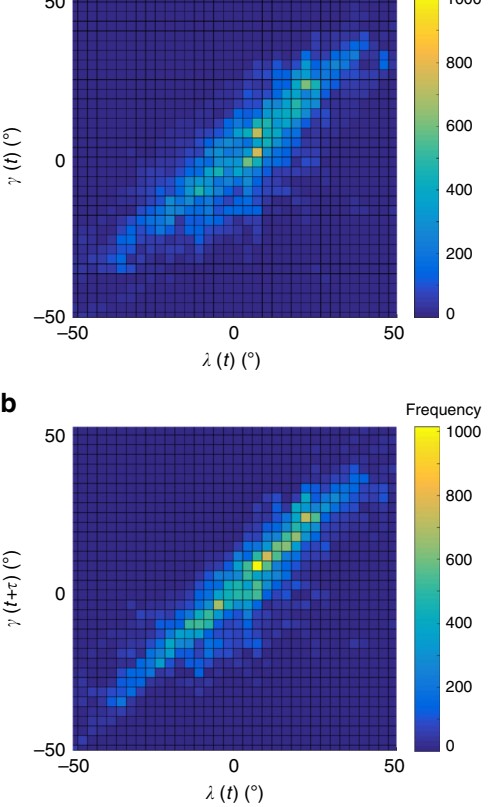

**Fig. 3** Correlation of track angle and line-of-sight angle. Each figure panel plots a two-dimensional histogram of track angle $\gamma$ against line-of-sight angle $\lambda$ for all sample points over all flights: **a** no delay between $\gamma(t)$ and $\lambda(t)$; **b** track angle $\gamma$ lagged by $\tau = 0.16$ s delay, representing the median value of the delay at which the correlation between $\gamma(t + \tau)$ and $\lambda(t)$ was maximised for each flight. The linear association in **b** indicates that the attack trajectories of the hawks approximated a delayed pure pursuit course

delay $\tau$ and guidance constant $K$ or $N$ for each flight independently, minimising the prediction error $\varepsilon$, defined as the mean absolute distance between the measured and simulated trajectories. To eliminate the attendant risk of overfitting, we then tried fitting the parameters $\tau$, $K$ and $N$ under the mixed guidance law to all flights together, holding their values the same across all 50 flights. All three guidance laws were capable of simulating the majority of the flight trajectories closely, when fitting their guidance parameters independently to each flight. Beginning with the two simplest guidance laws with only two fitted parameters per flight, the median prediction error was $\tilde{\varepsilon} = 0.46$ m for PP (IQR: $0.93 - 0.26$ m; Bootstrapped 95% CI: 0.35, 0.70 m) and $\tilde{\varepsilon} = 0.51$ m for PN (IQR: $0.96 - 0.19$ m; Bootstrapped 95% CI: 0.32, 0.72 m), or just over 1% of the median distance flown (Supplementary Table 2; Fig. 5a, d).

The 50 independently fitted values of $N$ followed a symmetric and well-behaved distribution (Fig. 5e), with a clear mode at $N \approx 1$ as expected if the hawks had used PN to implement pursuit ($\tilde{N} = 0.9$; IQR: $1.3 - 0.7$). In contrast, the 50 independently fitted values of $K$ were highly skewed, with no clear mode ($\tilde{\tau} = 0.00$ s$^{-1}$: IQR: $6.1 - 1.1$ s$^{-1}$; Fig. 5b) and some extreme outliers (Supplementary Table 2). The mean ranks of the independently fitted values of $N$ varied significantly between individuals (Kruskal–Wallis test: $\chi^2(4) = 12.96$, $p = 0.01$), but a

post hoc test found evidence of only one significant pairwise difference between birds, so we do not attribute much importance to this result. There was no evidence of any significant variation in the independently fitted values of $K$ between individuals (Kruskal–Wallis test: $\chi^2(4) = 3.16$, $p = 0.53$). Interestingly, the majority of the PP simulations entailed an effectively instantaneous response ($\tilde{\tau} = 0.00$ s; IQR: $0.09 - 0.00$ s; Fig. 5c), which in practice would imply the presence of a predictive element to overcome the inevitable sensorimotor delay. In contrast, the PN simulations typically involved a more delayed response, with a median fitted delay of $\tilde{\tau} = 0.10$ s (IQR: $0.19 - 0.00$ s; Fig. 5f), which is of similar magnitude to the sensorimotor delay identified in previous studies of avian guidance behaviours[10].

As most of the trajectories were well-modelled by either PP or PN, we next asked whether these elements might be combined in a mixed guidance law. Because the mixed guidance law reduces to PP in the special case that $N = 0$, and reduces to PN in the special case that $K = 0$, it will inevitably model the data at least as closely as either PP or PN if its guidance parameters are fitted to each flight independently. When fitting this mixed guidance law independently to each flight, the median prediction error was indeed significantly lower than for either PP or PN, at $\tilde{\varepsilon} = 0.18$ m (IQR: $0.36 - 0.09$ m; Bootstrapped 95% CI: 0.13, 0.25 m; Fig. 5g), or <0.5% of the median distance flown (Supplementary Table 2). Moreover, the populations of the independently fitted values of $N$ and $K$ under the mixed guidance law followed a tighter and more symmetric distribution ($\tilde{N} = 0.7$; IQR: $0.8 - 0.4$; $\tilde{K} = 1.0$ s$^{-1}$; IQR: $1.4 - 0.2$ s$^{-1}$; Fig. 5h) than they did under either PP or PN (Fig. 3b, e). Likewise, the median delay fitted under the mixed guidance law was close to the delay of ~0.1 s fitted under PN, but with a narrower spread ($\tilde{\tau} = 0.12$ s; IQR: $0.17 - 0.02$ s; Fig. 5i). The obvious difficulty with interpreting the results of these independently fitted models is that they all risk overfitting to a greater or lesser degree, with a total of 150 independently fitted parameters under the mixed guidance model, and 100 independently fitted parameters for each of PP and PN (see Supplementary Table 2).

To eliminate the risk of overfitting completely, we therefore searched for the unique combination of $N$, $K$ and $\tau$ that minimized the median prediction error under the mixed guidance law for all 50 flights simultaneously (Fig. 5j–l). This mixed guidance law with just three globally fitted parameters ($N = 0.7$, $K = 1.2$ s$^{-1}$, $\tau = 0.09$ s) had a median prediction error of $\tilde{\varepsilon} = 0.34$ m (bootstrapped 95% CI: 0.24, 0.53 m). Given that the independently fitted PP and PN guidance laws, with 100 model degrees of freedom each, had a higher median prediction error ($\tilde{\varepsilon} = 0.46$ m and $\tilde{\varepsilon} = 0.51$ m, respectively), it is therefore reasonable to prefer the globally fitted mixed guidance law with only 3 model degrees of freedom on grounds of parsimony, despite the fact that the 95% CIs on the median prediction error are overlapping. We conclude that the globally fitted mixed guidance law provides the best balance between goodness of fit and model parsimony amongst those we considered, closely capturing the observed turning behaviour on most of the flights with only three fitted parameters (Figs. 4, 5; Supplementary Figs. 1–9).

## Discussion

The sensorimotor feedback requirements of the PP and PN elements of this mixed guidance law are quite different. Specifically, the deviation angle $\delta$ that is fed back under PP is the angle between the velocity vector of the pursuer and its line-of-sight to target, and is therefore defined egocentrically. This angle could be estimated in various ways, but will be similar to the angle between the body axis and the sagittal plane if the head is assumed to be kept level and to track the target closely. In contrast, the

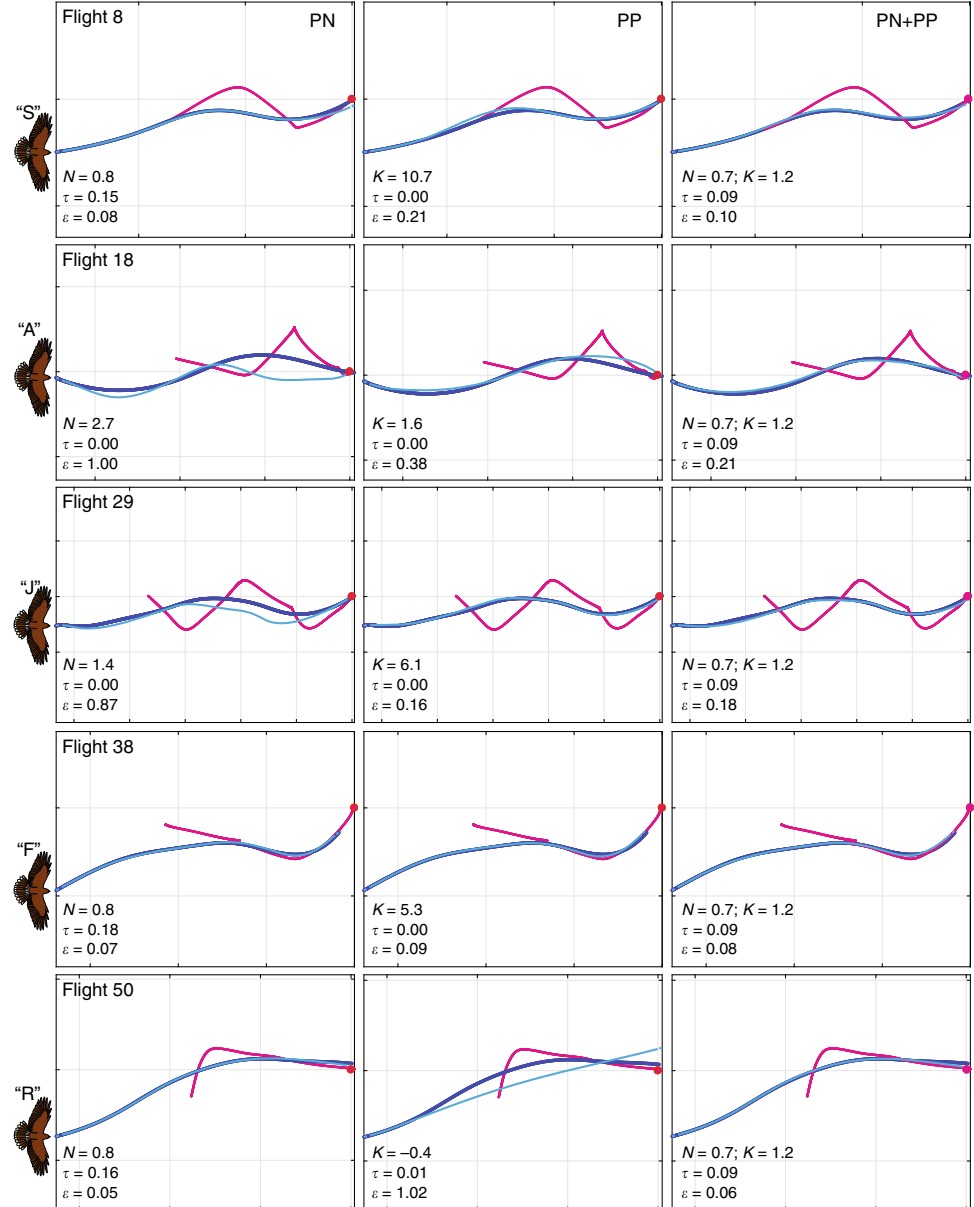

**Fig. 4** Comparison of measured and simulated attack trajectories. Panels display measured attack trajectories (dark blue) and best-fitting simulations (light blue) for each Harris' Hawk in pursuit of the lure (magenta), arranged by individual bird (rows) and by guidance law (columns; PN: proportional navigation; PP: proportional pursuit; PN + PP: mixed guidance law). The time delay $\tau$ (s) and guidance constant $K$ (s$^{-1}$) or $N$ are independently fitted to each flight for PP and PN, but are globally fitted to all flights for the mixed guidance law. For each bird, we display the flight with the least prediction error $\varepsilon$ (m) under the mixed guidance law (excluding flights <20 m), to best show the complementarity of its PN and PP elements: whilst all flights are well-modelled by the globally fitted mixed guidance law, some are not well-modelled by PN or PP, despite these being independently fitted to each flight; see Supplementary Figs. 1–9 for all other flights. Grid spacing: 10 m

line-of-sight rate $\dot{\lambda}$ that is fed back under PN is defined in an inertial frame of reference. Under the same set of assumptions on head tracking, the line-of-sight rate could be estimated either by integrating the angular accelerations sensed by the vestibular system, or by making direct use of the rotational optic flow cues produced by the head's self-motion relative to a fixed visual background. Feeding back the line-of-sight rate as well as the deviation angle under a mixed guidance law should improve the speed of response and reduce overshoot in a pursuit, because the line-of-sight rate provides a prediction of how the deviation angle is changing. This is consistent with the observation that our best-fitting PP simulations entailed effectively instantaneous sensory feedback (Fig. 3c) requiring some form of prediction, in

contrast to the realistically delayed feedback that we found in the simulations fitted under PN (Fig. 3f) or the mixed guidance law (Fig. 3i–k).

Why, though, should hawks tune their mixed guidance law to produce a trajectory approaching a pure pursuit, rather than a parallel navigation course? Whereas the intercept trajectories associated with parallel navigation are time-optimal against non-manoeuvring targets[7], and may be nearly time-optimal against manoeuvring ones[14], the tail-chase trajectories associated with pure pursuit are usually thought to be energetically costly and inefficient[15]. It is an open question how these different kinds of attack behaviours might perform in response to closed-loop evasive manoeuvres. Pure pursuit has been observed in predatory

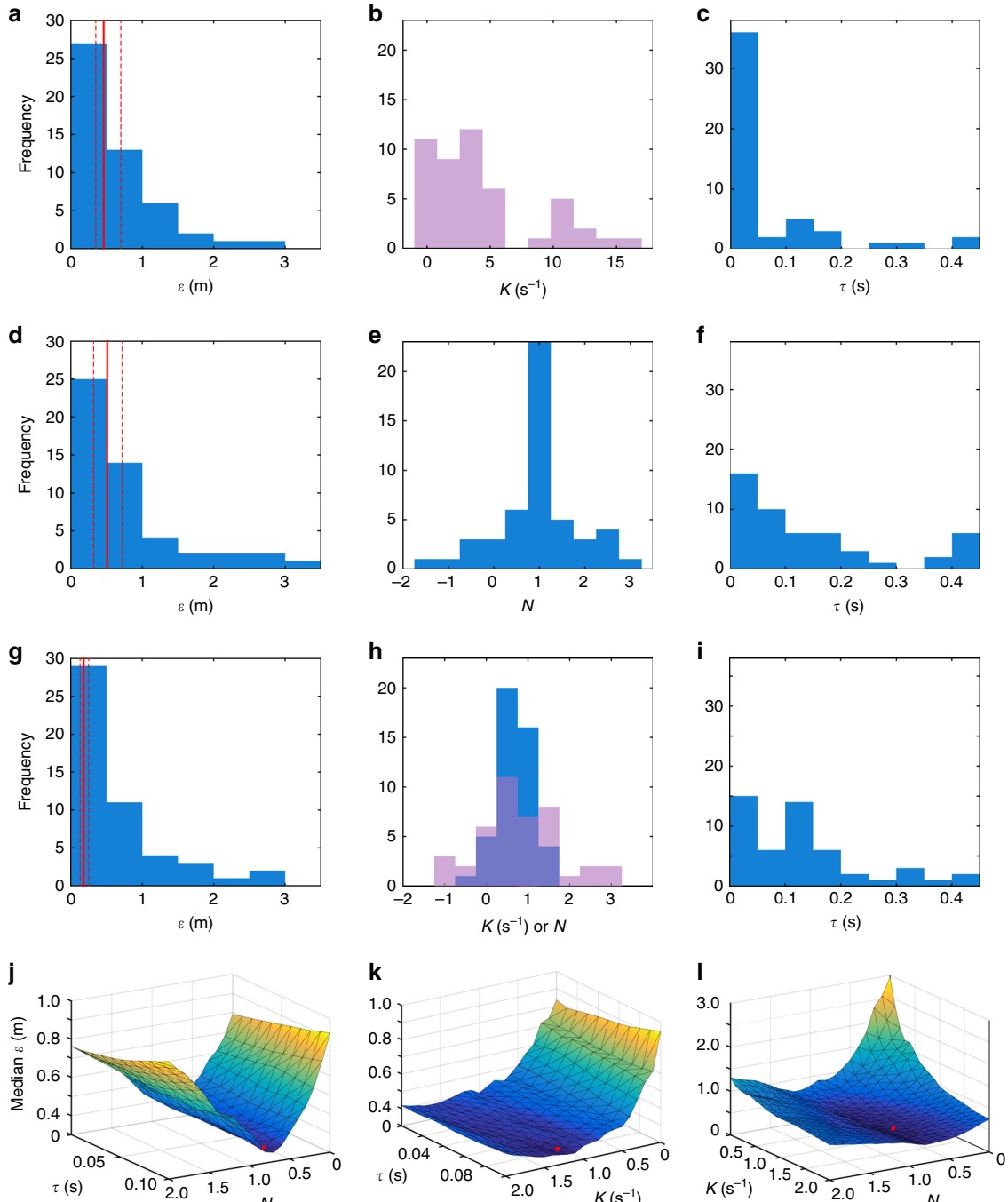

**Fig. 5** Model parameters and fit under each guidance law. **a–c** proportional pursuit (PP), fitted independently to each flight; **d–f** proportional navigation (PN), fitted independently to each flight; **g–i** mixed (PP + PN) guidance law, fitted independently to each flight; **j–l** mixed (PP + PN) guidance law, fitted globally to all flights. **a**, **d**, **g** histograms showing the prediction error $\varepsilon$ of the best-fitting model for each flight, where the solid and dashed red lines denote the median prediction error $\tilde{\varepsilon}$ and its bootstrapped 95% confidence interval; **b**, **e**, **h** histograms showing the best-fitting guidance constant $K$ (magenta) and/or $N$ (blue) for each flight; **c**, **f**, **i** histograms showing the best-fitting delay $\tau$ for each flight. **j–l** Surface plots showing how median prediction error $\tilde{\varepsilon}$ for all flights varies as a function of each pair of the parameters $N$, $K$ and $\tau$ of the mixed guidance law, holding the third parameter constant at its best-fitting value. Surface colour denotes surface height; red dot denotes location of global optimum

cursorial beetles[16,17], but other aerial predators including dragonflies[18,19], flies[15,20], bats[14] and falcons[6,21] approximate a parallel navigation course. Previous research on another hawk species found a mix of both[8], which might be explained by the use of a mixed guidance law combining elements of PP and PN. To understand the basis of this interspecific variation, we must take a comparative approach to the dynamics. Conveniently, whilst the globally fitted mixed guidance law provides the more

parsimonious and marginally better-fitting model, a PN guidance law is capable of describing our hawks' measured attack trajectories nearly as closely if its parameters are fitted independently for each flight, enabling phenomenological comparison of their fitted $N$-values with those of falcons (Fig. 6a).

Whereas our Harris' Hawks attacked jinking ground targets at $N \approx 1$ (bootstrapped 95% CI for $\tilde{N}$: 0.81, 1.01), Peregrine Falcons have been found to operate at significantly higher $N$-values

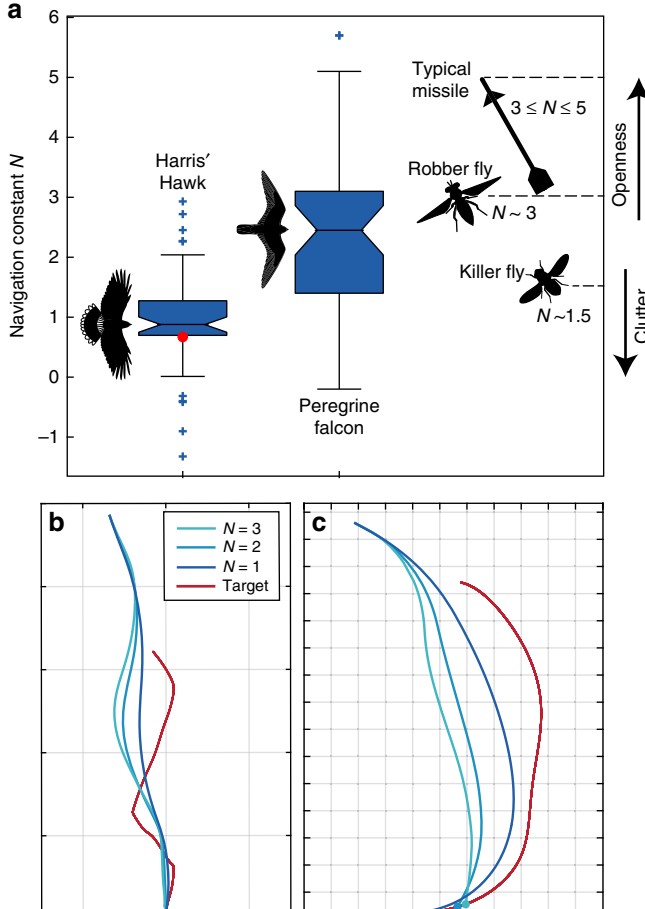

**Fig. 6** Comparative analysis of guidance behaviour. **a** Box-and-whisker plots comparing $N$-values for PN guidance models fitted independently to 50 flights from Harris' Hawks (this paper) and 42 flights from Peregrine Falcons[5]. Centre line of box denotes median; bounds of box denote first and third quartiles. Crosses indicate outlying points falling >1.5 times interquartile range below the first quartile or above the third quartile; whiskers extend to the most extreme data points not considered outliers. Red dot denotes $N$-value for globally fitted mixed guidance law. Silhouettes show typical values for missiles[6] and predatory flies[15,20]. **b**, **c** Trajectories simulated under PN guidance at $N \in \{1, 2, 3\}$, in pursuit of (**b**) jinking ground target from hawk experiments, and (**c**) aerial target from falcon experiments. Initial position of attacker as measured; initial track angle directed at target to avoid bias. Coloured circles denote collision. Grid spacing: 10 m. Fly icons redrawn from original artwork in refs. [15,20] under CC BY 4.0

against stationary and manoeuvring targets mimicking typical prey behaviours (bootstrapped 95% CI for $\tilde{N}$: 1.76, 2.87). For falcons attacking stationary ground targets[6], the median $\tilde{N} = 2.6$ is remarkably close to the well-known theoretical optimum of $N = 3$ for this case[7]. The same median was also observed in falcons attacking manoeuvring aerial targets[6], which is interesting because a physics-based simulation study[3] has found that the catch success of model falcons against erratically manoeuvring prey is maximized at $N \approx 3$. Hence, the guidance system of Peregrine Falcons does appear to be optimised in relation to their flight ecology. Is the same true of Harris' Hawks? To answer this, we simulated the effect of parametrically varying $N \in \{1, 2, 3\}$ in attacks on targets drawn from our experiments with hawks and falcons. Intriguingly, capture of the jinking target occurred soonest at $N = 1$ (Fig. 6b), whereas capture of the gently

manoeuvring target used with falcons occurred soonest at $N = 3$ (Fig. 6c). For these examples, at least, the different $N$-values observed in hawks and falcons therefore work best on the kinds of targets against which they were observed. More generally, lower $N$-values cause lower amplification of a jinking target's twists and turns, reducing the extent to which these throw the attacker off course (Fig. 6b). This is especially important at close range, as the angular effect of an evasive motion declines with distance. The guidance behaviour of Harris' Hawks is therefore appropriately tuned for close pursuit of the terrestrial prey on which they specialise.

We did not directly study the role of habitat clutter in this experiment, but the clutter that is typical of the habitats in which hawks hunt offers another possible functional account of why a mixed guidance law might be advantageous. The intercept trajectories produced at high $N$-values work well in the open, but need not result in a feasible path through clutter. In contrast, the tail-chase trajectories produced at low $N$-values, which the PP element of a mixed guidance law reinforces, inherently cause a pursuer to follow the lead of its prey through clutter. The same reasoning might also explain the behaviour of predatory flies: the robber fly *Holcocephala fusca* intercepts prey in the open and operates at $N \approx 3$ like a falcon, whereas the killer fly *Coenosia attenuata* hunts in clutter and operates at a lower value of $N \approx 1.5$, more like a hawk[15]. Hence, across two phyla and five orders of magnitude of body mass, two aerial predators that intercept prey in the open do so much like the guided missiles they resemble, whereas two that closely pursue prey in clutter operate at lower feedback gain (Fig. 6a). We therefore predict that, when comparing attack trajectories phenomenologically, low $N$-values will typify aerial, aquatic and cursorial predators that pursue agile prey through clutter, whereas higher values of $N \approx 3$ will typify predators that intercept their prey in open environments.

## Methods

**Experimental design.** We flew $n = 5$ captive-bred Harris' Hawks *Parabuteo unicinctus* (Supplementary Table 1), at an unpredictably manoeuvring target simulating the jinking manoeuvres of a typical prey item (Fig. 7). Tests with birds R and F were conducted on a sloping open grassy field near Abergavenny, UK, from October to December 2012; tests with birds A, J and S were conducted on another sloping open grassy field in the same area, from July to September 2013. Each pursuit was filmed using four S-PRI high-speed video cameras (Lake Image Systems Ltd, Tring, UK) fitted with 28 mm f/2.8D lenses (Nikon Corporation, Tokyo, Japan), recording RGB footage at 250 Hz (1280 × 1024 pixels; 0.002 s exposure). The cameras were manually post-triggered to give 8 s recording time, and were calibrated as detailed below to enable three-dimensional (3D) reconstruction of the trajectory of the hawk and the lure (see Videogrammetry). We continued testing until we had obtained ≥20 flights for each bird, in which the lure was intercepted successfully within view of the cameras. We recorded many other flights in which the hawk failed to intercept the lure, but we do not analyse them here because it is uncertain whether the hawk was locked on to its target for the duration of those flights. To deal with the volume of data (~1 M frames in total), and to ensure a balanced design, we determined to analyse only the last 10 flights from each individual. All individuals therefore experienced a training period of ≥10 successful flights before the set of flights that we analysed.

**Experimental protocol.** The hawks were fitted with a falconry harness (TrackPack, Marshall Radio Telemetry, North Salt Lake, UT, USA), comprising a plastic mounting plate held between the shoulders by a pair of Teflon ribbons drawn once around the bird's body in a figure-of-eight. The harness was fitted with two brightly coloured 0.04-m-diameter polystyrene balls attached to the crossover point on the breast and mounting plate on the back. The hawks were flown individually from the top of the test area, taking off at will, as soon as the lure started moving, from either a T-shaped perch, the falconer's fist, or a low tree branch. A 0.1 m diameter food lure was attached by 100 m of kite line to an electric winch (Expert Winch, Gliders Distribution, Newark, UK) that pulled the lure at speed down a zigzagging course through a mown grassy field with a 1:10 slope (Fig. 7). The lure line was run around a series of pulleys to generate abrupt changes in direction. We used ten pulleys in total, six of which were built into tunnels to guide the lure and motivate the hawks to chase the intermittently vanishing target. This allowed the lure to be dragged along 16 alternative courses from three different starting positions, switching direction four to six times on each course (Fig. 7). We randomly selected

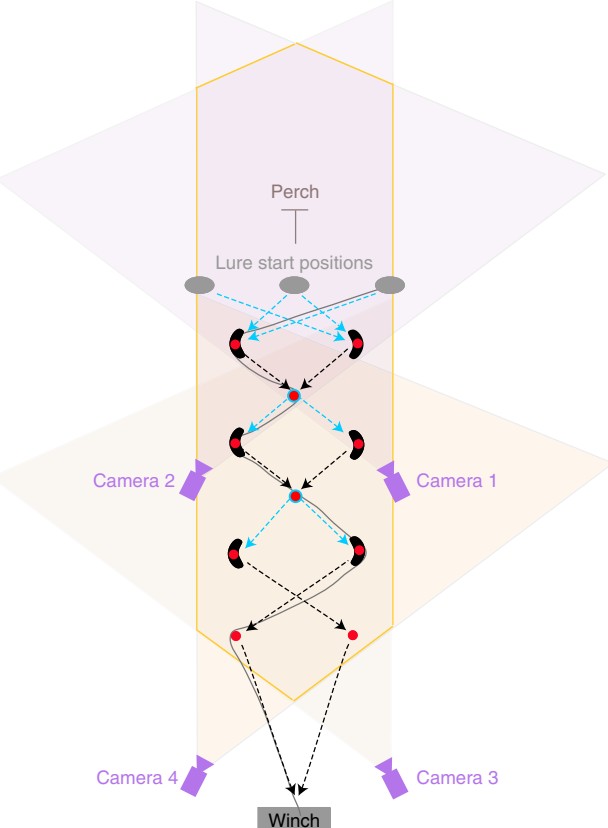

**Fig. 7** Schematic of experimental design. Overhead view, with shaded wedges denoting the overlapping fields of view of the cameras. The bird began its attack from a perch positioned behind the starting position of the lure (not to scale), which followed a zigzagging course around a series of pulleys (red circles). The lure began moving from one of three covered start positions (grey ovals), and was then pulled around a randomised subset of six from a total of ten pulleys. The lure was always drawn around the two central pulleys (circled blue), but the direction of its motion away from these pulleys was made unpredictable by randomising which of the left or right outer pulleys the lure was drawn towards (blue arrows). Outer pulleys were covered by tunnels (black curves), to guide the lure and to motivate chasing behaviour. The experimenter attempted to keep the lure ahead of the hawk until the end of the course, by controlling the speed of the lure. The lure's start position was randomised on each trial, as was the subset of outer pulleys around which the lure was pulled, giving a total of 16 possible courses for the lure to follow. Dummy lines were laid to avoid the bird reading the course that had been laid ahead of the flight. An illustrative lure trajectory is shown as a grey line. Blue dashed arrows denote sections where the lure's direction of travel was unpredictable; black dashed arrows denote sections of the lure's trajectory where the direction of travel might have been anticipated if the bird had learned that the lure was always pulled towards the central pulleys from the outer pulleys

which course to use on each flight, laying dummy lines and placing covers over the starting positions to prevent the birds anticipating the course. The lure was always pulled towards the middle of the course at the outside pulleys, so it is possible in principle that the hawks could have learned to anticipate the direction of these switches. The lure could be pulled in either direction at the middle pulleys, so the birds could not have learned to anticipate the direction of these switches, unless by discriminating which line was under tension. The birds were not motivated to continue chasing the lure if it was allowed to outpace them, so the winch was controlled manually and stopped at the point of capture to avoid harming the bird.

**Videogrammetry.** The cameras were calibrated on every test day by filming a calibration object that was moved through the test volume in a range of

orientations. The calibration object comprised a 1.05-m long clear acrylic tube with a coloured marker ball at each end. The pixel coordinates of the two marker balls were tracked automatically (see below), and a self-calibrating bundle adjustment was used to identify jointly optimal estimates of the camera extrinsic and intrinsic parameters and the pose of the calibration object, by solving the camera collinearity equations using a nonlinear least squares solver in Matlab (The Mathworks Inc., Natick, MA). Because lens distortions were minimal, we assumed a simple perspective projection. Field measurements of camera position were used to specify initial estimates of the extrinsic parameters to ensure convergence of the solver. Having first optimized the intrinsic parameters separately for each flight, we ultimately fixed the principal distance of the cameras at 28.4 mm for all flights, calculated as the mean for all cameras over all flights. This was done on the basis that there was no systematic variation in the estimated principal distance between cameras, so that the grand mean represents the best estimate of the true value of this parameter.

We established the error of our camera calibration by calculating the standard deviation of the estimated distance between the end markers of the calibration object over all of the frames. The camera calibration procedure treats these markers as fixed points on a rigid body at 1.05 m spacing. Having calibrated the cameras under this assumption, we then relaxed this rigid body constraint by estimating the positions of the end markers independently, using the same code that was subsequently used to estimate the position of the lure and the markers on the bird. Because the calibration object was moved through the test volume in a range of different positions and orientations, the standard deviation of the estimated distance between its end markers provides an isotropic estimate of the measurement error. Across trials, the standard deviation of the estimated distance between the calibration object end markers was 0.023 m, with 95% of the deviations from the mean falling on the interval $[-0.032, 0.036]$ m. It follows that the typical error of our camera calibration is approximately three orders of magnitude smaller than the length of the measurement volume.

The pixel coordinates of the lure and the markers on the bird were automatically identified using custom-written software in Matlab (The Mathworks Inc., Natick, MA). Tracking of the lure and markers was done automatically using custom-written code in Matlab. The user first created a colour template by manually identifying the tracked object in a series of specified video frames. A local search area was then defined around the marker, and the software used the colour template to identify matching pixels within this region of interest, updating the search area by centring it on the last successfully tracked point. To identify matching pixels, the colour of each pixel within the search area was compared with the distribution of the colour template by computing its Mahalanobis distance. Any pixels with a Mahalanobis distance below a set threshold were treated as candidate marker pixels, and the centroid of the single largest contiguous group of candidate marker pixels was assumed to correspond to the centroid of the marker itself. The software used a Kalman filter to estimate the position of the marker if no marker was found, or if the estimated position of the marker had moved too far between frames. Frames in which the markers were not visible were treated as missing data. All estimated marker positions were checked by the user and corrected manually if necessary.

We usually tracked only the bird's breast marker, but used the back marker for the whole flight if the breast marker was obscured. The trajectories that we recorded were always close to planar, and we therefore projected them onto the ground plane defined by the first and second principal components of the lure's path. All subsequent analysis was performed on this two-dimensional (2D) projection of the data, which always captured >98% (typically >99.9%) of the 3D variation in the bird's measured position (Supplementary Table 2). We defined the track angle of the bird ($\gamma$) as the polar angle of its velocity vector in the ground plane, and the line-of-sight angle ($\lambda$) as the polar angle of the vector from bird to lure. We used cubic interpolation to fill in any missing data points, before smoothing the 2D trajectories using quintic splines fitted at a tolerance designed to remove a root mean square error matching the diameter of the marker or lure. Finally, we differentiated and evaluated the splines analytically to estimate the velocity and acceleration of the bird and lure at 16 kHz, which ensured an adequately small integration step size for our simulations.

**Simulations of guidance behaviour.** We simulated the hawks' measured flight trajectories by predicting the hawk's turning rate $\dot{\gamma}(t)$ using (i) a proportional pursuit (PP) guidance law $\dot{\gamma}(t) = -K\delta(t - \tau)$, where $\delta = \gamma - \lambda$ is the deviation angle and $K$ is a guidance constant; (ii) a proportional navigation (PN) guidance law $\dot{\gamma}(t) = N\dot{\lambda}(t - \tau)$, where $\dot{\lambda}(t)$ is the line-of-sight rate and $N$ is a guidance constant called the navigation constant; (iii) a mixed guidance law combining PP and PN elements to command turning as $\dot{\gamma}(t) = N\dot{\lambda}(t - \tau) - K\delta(t - \tau)$. We used the same simulation algorithm as in our previous work on falcons[6], given knowledge only of the hawk's initial track angle and initial position, and given the complete time history of the lure's motion, always matching the hawk's simulated groundspeed to its measured groundspeed (see Supplementary Data 1 for raw trajectory data and simulation code).

Writing the simulated position vector of the pursuer as $\mathbf{x}_P$, and the measured position vector of the target as $\hat{\mathbf{x}}_T$, we define the line-of-sight vector ($\mathbf{r}$) as:

$$\mathbf{r} = \hat{\mathbf{x}}_T - \mathbf{x}_P \qquad (1)$$

and the closing velocity ($\mathbf{v}_C$) as:

$$\mathbf{v}_C = \mathbf{v}_P - \hat{\mathbf{v}}_T \qquad (2)$$

The deviation angle ($\delta$) is given in vector form as:

$$\boldsymbol{\delta} = \left( \cos^{-1} \frac{\mathbf{r} \cdot \mathbf{v}_P}{|\mathbf{r}| \, |\mathbf{v}_P|} \right) \left( \frac{\mathbf{r} \times \mathbf{v}_P}{|\mathbf{r} \times \mathbf{v}_P|} \right) \qquad (3)$$

whilst the line-of-sight rate ($\omega$) is given in vector form as:

$$\boldsymbol{\omega} = \frac{\mathbf{r} \times (-\mathbf{v}_C)}{|\mathbf{r}|^2} \qquad (4)$$

Under PP, the pursuer's turning at time $t$ is commanded at a rate proportional to the deviation angle at time $t - \tau$, such that its commanded centripetal acceleration is given by:

$$\mathbf{a}_P(t) = -K\boldsymbol{\delta}(t - \tau) \times \mathbf{v}_P(t) \qquad (5)$$

where $K$ is a guidance constant. Under PN, the pursuer's turning is commanded at a rate proportional to the line-of-sight rate, such that its commanded centripetal acceleration is given by:

$$\mathbf{a}_P(t) = N\boldsymbol{\omega}(t - \tau) \times \mathbf{v}_P(t) \qquad (6)$$

where $N$ is a guidance constant called the navigation constant. Under the mixed guidance law, these PP and PN elements are combined to command the pursuer's centripetal acceleration as:

$$\mathbf{a}_P(t) = N\boldsymbol{\omega}(t - \tau) \times \mathbf{v}_P(t) - K\boldsymbol{\delta}(t - \tau) \times \mathbf{v}_P(t) \qquad (7)$$

In our simulations, the preceding equations are implemented in discrete time, and coupled by the difference equations:

$$\mathbf{x}_{P_{n+1}} = \mathbf{x}_{P_n} + \Delta t \, \mathbf{v}_{P_n} \qquad (8)$$

$$\mathbf{v}_{P_{n+1}} = \hat{v}_{P_{n+1}} \frac{\mathbf{v}_{P_n} + \Delta t \, \mathbf{a}_{P_n}}{|\mathbf{v}_{P_n} + \Delta t \, \mathbf{a}_{P_n}|} \qquad (9)$$

where $\hat{v}_P = |\hat{\mathbf{v}}_P|$ denotes the actual measured speed of the pursuer, and where the subscript notation indicates the values of the variables at successive time steps, such that $t_{n+1} = t_n + \Delta t$. The step size $\Delta t$ in our simulations was made small enough to guarantee the accuracy of the fitted guidance parameters and prediction error (see below) to the level of precision at which they are reported ($\Delta t = 6.25 \times 10^{-5}$ s). The first of these difference equations advances the simulated position of the pursuer from $\mathbf{x}_{Pn}$ to $\mathbf{x}_{Pn+1}$ using the simulated velocity $\mathbf{v}_{Pn}$. The second difference equation advances the simulated velocity of the attacker from $\mathbf{v}_{Pn}$ to $\mathbf{v}_{Pn+1}$ by using the commanded centripetal acceleration $\mathbf{a}_{Pn}$ to rotate $\mathbf{v}_{Pn}$, but then scales the rotated vector to match the measured speed of the attacker ($\hat{v}_{P_{n+1}}$). Code implementing these equations of motion is provided in Supplementary Data 1.

**System identification**. We fitted the guidance constant $K$ or $N$ and time delay $\tau$ under the PP and PN guidance laws for each flight independently, modelling every possible delay $\tau \leq \tau_{max}$ at a spacing corresponding to the 0.004 s inter-frame interval where $\tau_{max} = 0.4$ s. Taking time $t = 0$ as take-off, we simulated the bird's flight trajectory from $t = \tau_{max}$ to the point of intercept or first near-miss. This ensured that the same section of flight was analysed for all time delays. For each value of $\tau$, we used a Nelder-Mead simplex algorithm in MATLAB to find the value of $K$ or $N$ that minimised the prediction error $\varepsilon$, defined as the mean absolute distance between the measured ($\hat{\mathbf{x}}_P$) and simulated ($\mathbf{x}_P$) positions of the bird at each time point:

$$\varepsilon = \frac{1}{k} \sum_{n=1}^{k} |\mathbf{x}_{P_n} - \hat{\mathbf{x}}_{P_n}| \qquad (10)$$

We then optimized $\tau$ by selecting the value that minimised the prediction error $\varepsilon$ over all of the fitted values of $\tau$.

We fitted the mixed guidance law globally to all of the flights, identifying the unique combination of the guidance constants $K$ or $N$ and the time delay $\tau$ that minimised the median prediction error $\tilde{\varepsilon}$ over all flights. This was done through an exhaustive search of the prediction error $\varepsilon$ for values of $K$ and $N$ on the interval [0, 2] at 0.1 spacing, and values of $\tau$ on the interval [0, 0.1] at 0.004 s spacing, these values being chosen in light of the independently fitted optima for PP and PN. Finally, to improve the precision of the estimates, we refined the search by a factor of 10 in the vicinity of the identified optimum.

**Statistical analysis**. To test whether the hawks' trajectories followed a pure pursuit course, we compared the track angle $\gamma(t)$ at time $t$ with the angle of the line-of-sight to target $\lambda(t - \tau)$ at delay $\tau$, by binning all of the data in a two-dimensional (2D) histogram. We also computed the cross-correlation sequence of $\gamma$ and $\lambda$ independently for each flight. We analysed the statistical distributions of the fitted guidance parameters and prediction error across all 50 flights, reporting their median (denoted by a tilde) and interquartile range (IQR) for robustness against outliers and skew. Where relevant, we also report bootstrapped 95% confidence intervals (CIs) for the medians. These were computed using the bias-corrected and accelerated percentile method over $10^6$ iterations, using a customised version of the

corresponding function in MATLAB to implement multistage resampling by individual as appropriate to the structure of the data. The hierarchical structure of the published data on Peregrine Falcons[4] was too complicated to handle by this method, so each attack pass had to be treated as independent when bootstrapping these data. A two-tailed Kruskal–Wallis test was used to test for differences in the mean ranks of the fitted guidance constants between individuals, with post hoc testing done using Tukey's HSD test at $\alpha = 0.05$.

**Ethics statement**. We affirm that we have complied with all relevant ethical regulations for animal testing and research. The study protocol was approved by the United States Air Force, Surgeon General's Human and Animal Research Panel, and by the Local Ethical Review Committee of the University of Oxford's Department of Zoology, and was considered not to pose any significant risk of causing pain, suffering, damage or lasting harm to the animals.

**Reporting summary**. Further information on research design is available in the Nature Research Reporting Summary linked to this article.

## Data availability
All of the high-speed video datasets generated and analysed during the current study are available from the corresponding author on reasonable request. All of the reconstructed trajectory data analysed during this study are included in the Supplementary Information File Supplementary Data 1 accompanying this published article.

## Code availability
The custom code used to simulate attack trajectories in the current study is included in the Supplementary Information File Supplementary Data 1 accompanying this published article.

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

## Acknowledgements

We thank our falconer Martin Cray, and Jo Binns for animal husbandry and land access. We thank Shane Windsor for tracking code, Simon Walker for calibration code, and Marco Klein Heerenbrink for creating the Harris' Hawk icon used in the figures. We thank Robin Mills and Adrian Thomas for many useful discussions, and Chris Hudson for helpful suggestions. Effort sponsored by the Air Force Office of Scientific Research, Air Force Materiel Command, USAF, under grant number FA8655-11-1-3065. This project has received funding from the European Research Council (ERC) under the European Union's Horizon 2020 research and innovation programme (Grant Agreement No. 682501).

## Author contributions

G.T. conceptualized study; C.B. performed experiments; G.T. and C.B. designed experiments, performed analyses, and wrote paper.

## Additional information

**Competing interests:** Authors declare no competing interests.

