## [Peer Review File · Nature Communications]

Reviewers' comments:

Reviewer #1 (Remarks to the Author):

The authors present results from an extremely challenging empirical study of how Harris' Hawks chase an erratically moving target, and interpret their results using sophisticated computer simulations. I appreciate their clear and thorough presentation of the three-dimensional video tracking methods used to measure hawk and target trajectories and the many replicates used (5 different birds, 50 distinct trajectories). The empirical side of this study alone is a heroic undertaking and merits eventual publication in Nature Communications.

One of the difficult aspects of this field is the need to invent new methods of data analysis in addition to recording complex data under demanding conditions. The authors have been pioneers in inventing new ways to study navigational guidance in animals, in combination with other research groups cited in the references. In particular, they have introduced rigorous methods for studying proportional navigation in birds. Here they provide their MATLAB analysis code so others can replicate their modeling and simulations methods in other contexts. This will garner significant attention for this article once it is published.

This study fits empirical trajectories to computer simulations of three possible feed-back-based navigation law models: pure pursuit, proportional navigation and a combination of the two (mixed model). Each of these analyses is convincing on its own, well-motivated by the literature, and well-matched to the empirical data in that each does a good job reproducing the measured trajectories with low median positional error.

Therefore, after important revisions indicated below are addressed, this work will meet the Nature Communications threshold of "novel and important research" "of high-quality and of interest to the specific research community" and represents "represent important advances within specific scientific disciplines".

Several of these revisions require changing the stated results from the computer simulations in important ways. None of these changes would diminish the impact of the research. All of these changes can be made without taking additional data, although some involve additional analysis. Any required changes to the text should fit easily within the tight word count of the journal. While they are stated at length, each should be addressable with minimal effort. (Though I do recommend some additional analysis that would enhance the scientific impact.)

I therefore recommend the authors be invited to “revise their manuscript to address specific concerns before a final decision is reached”. The revisions necessary to make the manuscript “technically sound” and to provide “strong evidence for its conclusions” are listed in approximate order of priority below.

Major revisions: The main conclusion of the paper should be reworded to agree with the methods and reported results (1-4) and to acknowledge very similar past work on hawks using similar models (5).

1) When taken as a whole, the results of their modeling do not currently support the main finding that hawks use the proposed mixed guidance law:

Abstract, lines 15-17: “Hawks use a mixed guidance law combining low-gain proportional navigation with a proportional pursuit element that promotes tail-chasing through clutter and is not thrown off by evasive maneuvers.”

Lines 110-111: the mixed model is both “best-fitting and most parsimonious”

Lines 138-139: “...the mixed guidance law provides the best model of our hawk trajectories...”

Actually, the three models studied gave the same median prediction error between simulated and measured trajectories at the 95% CI level (lines 100-102):

Model PP: pure pursuit [0.35, 0.70m]

Model PN: proportional pursuit [0.32, 0.72m]

Mixed model (PP + PN): [0.24, 0.53m]

While no p-values are given to support a finding that the mixed model has the lowest error of the three models considered, the extensive overlap between these confidence intervals suggests values well over $p = 0.05$.

Furthermore, Fig. 1G shows that the hawk’s velocity on average points toward the prey’s position once a reaction time lag is accounted for; this agreement is the definition of pure pursuit. (It would be interesting to know how this compared to measured reaction times for birds.) In addition, Fig.

3A,C show very similar histograms of the error for the pure pursuit and proportional navigation, although there is no similar data for the mixed model.

Thus, there is no statistical support provided for preferring the more complicated mixed model over pure pursuit, or for distinguishing these three models. One might reasonably state these results as supporting the simplest model, which is pure pursuit, as in lines 128-129: "...hawks tune their mixed guidance law to produce a trajectory approaching a pure pursuit...".

Thus, the conclusion should be restated as finding that all three models perform equally well at fitting the measured data. This study remains equally interesting, relevant and well-conducted if it supports the hawks studied using pure pursuit. There also may be a case that the mixed model is preferred as more biologically realistic, but the current manuscript does not make this argument. Possible additional data analysis (see below) might provide support for the mixed model or other alternatives. While this further analysis should not be required, I think it would strengthen the scientific argument.

The authors could determine whether the mixed model is a better fit to more complex target trajectories by fitting each trajectory, as opposed to globally fitting all 50 at once to avoid over-fitting (line 97), using a measure of goodness-of-fit that accounts for the different number of fit degrees of freedom in each model. The analysis also could determine whether median fit error correlates with path complexity. For example, median fit error could be plotted and correlated with the ratio of distance traveled to start-to-end distance, number of turns, etc.

2) On a related note, the manuscript states that parallel navigation has been ruled out, so the results of fitting the data using parallel navigation (and possibly a mixed parallel navigation and pure pursuit model) should be included:

lines 128-129: "Why, though, should hawks tune their mixed guidance law to produce a trajectory approaching a pure pursuit, rather than a parallel navigation course?"

Parallel navigation is not merely "geometric" (line 137): it is widely observed across many taxa and well-justified based on known mechanisms of visual processing and other sensory modalities (e.g. echolocation), as explained in many of the references cited. For example, a moving organism can implement a form of feedback-based parallel navigation by directly sensing optical flow of the target and making course adjustments to stabilize its visual angle.

Assuming that parallel navigation was tested as a possible model but the results not reported, the authors could simply add the line-of-sight lines to their plotted trajectories or provide an equivalent measure, in keeping with all the refs cited that studied pursuit in animals (12-19).

3) The manuscript implies that hawks were filmed chasing live “agile prey”, but the hawks studied chased only an artificial target (a box filled with food):

Title: “Hawks steer attacks using a guidance system tuned for close pursuit of agile prey through clutter”

Abstract, lines 14-15: “Here we identify the guidance law that Harris’ Hawks *Parabuteo unicinctus* use to chase jinking terrestrial prey in clutter.”

The manuscript should be revised to use a more accurate term such as “unpredictably-maneuvering target/dummy prey/falconry lure/etc.”, as in refs. (13), (15). Also, “jinking” is jargon unfamiliar to most readers of *Nature* and should be defined at first use.

These results are extremely interesting and relevant given this study design. There is much to be said for using a dummy target: it allows for reproducibility and replicability, varying trajectories on demand, and filming chases in a controlled location for imaging in three dimensions. Many other studies of animal navigation use them for this reason; this is especially suitable for organisms with poor visual acuity that may be unable to distinguish the difference from live prey, but this is not the case for birds.

However, because the target can influence the guidance law used (see below), the title, abstract and main text should describe the targets accurately, and note any related limitations when extrapolating the results to other contexts.

The target of pursuit has been shown to influence the guidance law used in other systems. E.g., bats chasing prey use parallel navigation (ref (12)) to pursue live prey and pure pursuit to chase another bat (e.g., Chiu, Chen, et al. "Effects of competitive prey capture on flight behavior and sonar beam pattern in paired big brown bats, *Eptesicus fuscus*." *Journal of Experimental Biology* 213.19 (2010): 3348-3356.)

Target motion should influence the choice of guidance law, as explained in the text. However, it does not explain how closely the lure’s motion resembles that of relevant prey. On video, the target looks

like a tumbling box. Surely a visually-oriented hawk is aware it is not chasing a live animal. How did the lure speeds, frequency and angles of maneuvering compare to those used by jackrabbits (the prey species of interest) during actual chases? The lure track diagram and videos make it seem like the lure changed directly relatively infrequently and at a limited number of known locations compared to escaping rabbits; the hawks underwent training on the same course before filming (lines 204-205); and target velocity changes did not occur in response to the hawk's motion, unlike the case for live prey. Lines 204-205, 217-227 in the methods also mention path predictability as a possible confounder. On a related note, earlier work not cited proposed a similar "integrated navigation law" in which the pursuer uses remembered landmarks to predict future motion using a combination of simpler guidance laws (e.g., Kong, Zhaodan, et al. "Perceptual modalities guiding bat flight in a native habitat." *Scientific reports* 6 (2016): 27252, Kong, Zhaodan, et al. "Optical flow sensing and the inverse perception problem for flying bats." *Decision and Control (CDC), 2013 IEEE 52nd Annual Conference on. IEEE, 2013; Kong, Zhaodan, et al. "Perception and steering control in paired bat flight." IFAC Proceedings Volumes 47.3 (2014): 5276-5282.)*

4) Although the title and abstract indicate that the hawks pursued target through clutter, the manuscript does not describe anything approximating clutter (e.g., brush, tall grass, branches, etc.) as being present during chases. Instead, the methods section (lines 215-217) describe the lure as moving across a "mown grassy field" guided by a series of winches and pulleys. This clutter-free environment also is depicted in the videos and images shown.

Either the paper should explain how clutter was included in the experimental design or else refrain from stating that chases in clutter were studied. Either way, it makes sense for the authors to discuss which navigation laws might work best in clutter in the context of their findings.

5) Ref. (6) proposed and provided evidence that hawks pursuing live maneuvering prey through cluttered natural environments use a mixed strategy (initial parallel navigation that transitions into pure pursuit at close quarters) during a large fraction of pursuits, and explicitly noted that proportional navigation might provide a more general model. This is so similar to the hypothesis tested in this study that it ought to be mentioned in the introduction. Instead, the introduction suggests that this research supported pure pursuit only:

lines 32-34: "Hawks may be expected to use a different guidance law, because their attacks often follow a pure pursuit course, in which flight is aimed directly at the target (6)."

and a more accurate summary is left for the discussion:

lines 135-136: "Previous research on another hawk species found a mix of both (6), which might be explained by the use of a mixed guidance law combining elements of PP and PN"

The revised version should credit this earlier work and other studies that use similar integrated/mixed navigation models, including the references cited at the end of comment 3 above and ref (18).

Minor revisions:

1) Harris' Hawks are pack hunters in their native environments. They presumably have evolved to solve a very different problem from other raptors, all of which hunt solo in most cases. How might this unique cooperative hunting strategy influence the navigation law they use for chasing prey? (Note that this point can be phrased to enhance the broader applicability of their findings.) This also merits a comparison with the results of studies of bats chasing prey cited above that found bats use parallel navigation to chase prey but pure pursuit to chase another bat during cooperative hunting. (See ref. (12) and Chiu, Chen, et al. cited above). Given that the Harris' Hawks use of pure pursuit, might this not arise from their similar use of cooperative hunting?

2) The methods section explains that the trajectories analyzed were in the ground plane (an appropriate choice here), but this is worth mentioning briefly in the main text or main figure captions.

3) The following sentences imply that the hawk's gaze-direction was measured:

line 48-50: "The hawks took off as soon as the lure began moving, flap-gliding to close range, and banking to turn whilst keeping their eyes level and their gaze directed at the target"

line 118: "under the assumption that the head tracks level and tracks the target closely"

The authors do not need to measure gaze-direction for this study. It seems unlikely gaze-direction can be determined from the videos, given the low resolution of the hawk's head and eyes, its varying visibility and the known complexity of hawk visual systems; head nystagmus is actually violated at the end of one video, although it's well established in the literature during level flight. The gaze-direction is not the same as the angle between the positions of the hawk's body and the lure (conventionally referred to as "line-of-site" in the literature; see line 281 in the methods).

These lines should be rephrase to avoid implying it was measured.

Suzanne Amador Kane

Reviewer #3 (Remarks to the Author):

The paper claims that Harris' hawk uses a combined proportional and pursuit based guidance strategy to intercept fleeing prey. The outcomes are convincing and I have made some suggestions below about clarifying some aspects. Appropriate references are made to the existing literature. This research is difficult to undertake and I believe the very best effort possible and practical has been made here, so I would not recommend any new or different experimental work. The statistical analysis is convincing and the results are quite clear and method reproducible by those with access to the same resources.

There is an early discussion of evolution of different raptor species, leading the reader to expect a strong hypothesis test related to evolution which does not happen. It is not wrong, but perhaps either more or less needs to be said.

In figure 1 The choice of lambda and gamma for key variables is unfortunate since it leads to visual confusion for many readers.

Starting line 53 Is the 0.16s sensorimotor delay based on a reasonable foundation? non-neuroscientists may be concerned about what could be seen as a free variable (I think it is a reasonable number for the several synapses and neuromuscular junction involved).

Line 66-68 it might be best to not say this regarding the missile guidance field because it does not accurately characterize what they are actually trying to achieve and the characteristics of their targets and where their interests lie. I don't think it would take anything away from the manuscript if it was not there.

Figure 3 has very flat error response with regard to time delay for all error surfaces, I notice in lines 90 to 91 a comment that the PP response was effectively instantaneous, which is not possible without some sort of prediction or tracking which in itself would then be part of the guidance law. This seems to be a complex situation with a simplistic statement to explain it.

At line 98 I don't think the word "Remarkably" adds any value.

102 the recognition that there is individual variability will immediately have readers wondering if the entire behaviour is learned from recent experience and the nature of the problem, or if the behaviour is hard-wired neural circuitry. A sophisticated vertebrate predator might have learned or at least tuned their innate behaviour to their environment.

Line 119 the vestibular system might be the source of the inertial reference frame, but there are visual possibilities here as well that seem more likely in a visual pursuit, or a fusion of both

For the discussion in the area of 130 to 140, it is worth pointing out the obvious that some of the prey species of vertebrate predators may be capable of sophisticated counter measures, a situation that is not tested in this work, since I assume the jinking of the target is open loop with regard to the hawk trajectory.

The conclusion area from 164 to 184 seems reasonable. I do wonder about headlining the drone issue as the last sentence.

For a lot of reasons drones are not a difficult target once detected and tracked, and are not likely to be for some time.

I believe this could be a classic work about animal pursuit. But I think there are stylistic elements that need to be worked on. The paper appears to be seeking justification for itself in other fields (missile guidance and counter-drone applications). This actually detracts from it being a classic work, since it might be criticized by those in other fields for aspects that are not really part of the hypothesis.

We thank both reviewers for their careful and positive reviews of our manuscript. We have made changes in response to all of their comments, to which a point-by-point response is given in red below. Besides making many small edits to the text, which are detailed below and highlighted in the tracked version of the manuscript, we have also made the following key changes: (i) we have included the inclusion of results from the independently-fitted mixed guidance law as suggested by Reviewer 1, including the addition of new figure panels; (ii) we have added a new figure plotting the lines-of-sight to show explicitly that the birds were not using parallel navigation; (iii) we have rearranged several of the figures, including bringing the schematic of the experimental design into the Methods section of the main text. We hope that these changes will address all of the reviewers' concerns.

Reviewer #1 (Remarks to the Author):

The authors present results from an extremely challenging empirical study of how Harris' Hawks chase an erratically moving target, and interpret their results using sophisticated computer simulations. I appreciate their clear and thorough presentation of the three-dimensional video tracking methods used to measure hawk and target trajectories and the many replicates used (5 different birds, 50 distinct trajectories). The empirical side of this study alone is a heroic undertaking and merits eventual publication in Nature Communications.

We are grateful to the reviewer for their thorough review and positive assessment of our manuscript.

One of the difficult aspects of this field is the need to invent new methods of data analysis in addition to recording complex data under demanding conditions. The authors have been pioneers in inventing new ways to study navigational guidance in animals, in combination with other research groups cited in the references. In particular, they have introduced rigorous methods for studying proportional navigation in birds. Here they provide their MATLAB analysis code so others can replicate their modeling and simulations methods in other contexts. This will garner significant attention for this article once it is published.

This study fits empirical trajectories to computer simulations of three possible feed-back-based navigation law models: pure pursuit, proportional navigation and a combination of the two (mixed model). Each of these analyses is convincing on its own, well-motivated by the literature, and well-matched to the empirical data in that each does a good job reproducing the measured trajectories with low median positional error.

Therefore, after important revisions indicated below are addressed, this work will meet the Nature Communications threshold of "novel and important research" "of high-quality and of interest to the specific research community" and represents "represent important advances within specific scientific disciplines".

Several of these revisions require changing the stated results from the computer simulations in important ways. None of these changes would diminish the impact of the research. All of these changes can be made without taking additional data, although some involve additional analysis. Any required changes to the text should fit easily within the tight word count of the journal. While they are stated at length, each should be addressable with minimal effort. (Though I do recommend some additional analysis that would enhance the scientific impact.)

I therefore recommend the authors be invited to "revise their manuscript to address specific concerns before a final decision is reached". The revisions necessary to make the manuscript "technically sound" and to provide "strong evidence for its conclusions" are listed in approximate order of priority below.

We thank the reviewer for their detailed and helpful comments, and hope they will agree that we have addressed these all through our revisions to the manuscript, and our point-by-point response below.

Major revisions: The main conclusion of the paper should be reworded to agree with the methods and reported results (1-4) and to acknowledge very similar past work on hawks using similar models (5).

We have revised the manuscript as suggested, and have detailed all of the changes in our responses to the specific points raised below.

1) When taken as a whole, the results of their modeling do not currently support the main finding that hawks use the proposed mixed guidance law:

Abstract, lines 15-17: "Hawks use a mixed guidance law combining low-gain proportional navigation with a proportional pursuit element that promotes tail-chasing through clutter and is not thrown off by evasive maneuvers."

Lines 110-111: the mixed model is both "best-fitting and most parsimonious"

Lines 138-139: "...the mixed guidance law provides the best model of our hawk trajectories..."

Actually, the three models studied gave the same median prediction error between simulated and measured trajectories at the 95% CI level (lines 100-102):

Model PP: pure pursuit [0.35, 0.70m]

Model PN: proportional pursuit [0.32, 0.72m]

Mixed model (PP + PN): [0.24, 0.53m]

While no p-values are given to support a finding that the mixed model has the lowest error of the three models considered, the extensive overlap between these confidence intervals suggests values well over $p = 0.05$.

We are confident that our conclusions are supported, but fully accept that the statistical basis of our inferences was not made sufficiently clear in the original manuscript. We have revised the manuscript extensively to address the reviewer's criticisms, and have explained these changes point by point below. In summary, whereas we initially presented the results of models fitted independently to each flight trajectory for proportional pursuit (PP) and proportional navigation (PN), we presented only the results of a global model fitted to all flight trajectories simultaneously for the mixed guidance law (PP+PN). This was done to guard against any risk that the PP+PN model was overfitted, noting that this mixed model contains PP and PN as special cases. We can see that this presentation was confusing, however, and have therefore added the results of the independently-fitted PP+PN models to the revised manuscript (new panels G-I in Fig. 5), which more clearly demonstrate the superiority of the mixed model.

The independently fitted PP+PN models fit the data do indeed fit the data much more closely than either PP or PN, with significantly lower median prediction error:

Independently-fitted PP models: median error = 0.46, 95% CI [0.35, 0.70m]

Independently-fitted PN models: median error = 0.51, 95% CI [0.32, 0.72m]

Independently-fitted PP+PN models: median error = 0.18, 95% CI [0.13, 0.25m]

It is evident, therefore, that the independently-fitted PP+PN models fit the data significantly more closely than do the independently-fitted PP or PN models, so that our main conclusion is sound. All of these models risk overfitting to a greater or lesser degree, with 100 or 150 fitted parameters, however, so we also present the results of the globally-fitted PP+PN model (3 parameters) as before.

The confidence interval for the median prediction error of the globally-fitted PP+PN model is:

Globally-fitted PP+PN model: median error = 0.34, 95% CI [0.24, 0.53m]

which does indeed overlap with the confidence interval for the independently fitted PP and PN models, as the reviewer notes. However, even if these models are treated as fitting the data equally well, the globally fitted PP+PN model is clearly to be preferred on grounds of parsimony, because it only 3 fitted parameters, as compared to the 100 fitted parameters for each of the independently-fitted PP and PN models. We think this is a compelling result, and are grateful to the reviewer for prompting us to clarify our logic. The relevant section now reads as follows:

“Given that the independently-fitted PP and PN guidance laws had a qualitatively worse median prediction error despite having 100 model degrees of freedom ($\tilde{\epsilon} = 0.46m$ and $\tilde{\epsilon} = 0.51m$, respectively, albeit that their 95% CIs overlap that of the global model), it is reasonable to favour the globally-fitted mixed guidance law with only 3 model degrees of freedom ($\tilde{\epsilon} = 0.34m$) on grounds of parsimony. We conclude that the globally-fitted mixed guidance law provides the best balance between goodness of fit and model parsimony amongst those we considered, closely capturing the observed turning behaviour on most of the flights with only 3 fitted parameters (Fig. 2; Figs. S2-S10).”

Furthermore, Fig. 1G shows that the hawk's velocity on average points toward the prey's position once a reaction time lag is accounted for; this agreement is the definition of pure pursuit. (It would be interesting to know how this compared to measured reaction times for birds.)

We agree, and had noted in the accompanying text that *“The correlation between $\gamma(t)$ and $\lambda(t - \tau)$ for each flight usually peaked at $r > 0.8$ with a small delay ($\tilde{\tau} = 0.16s$; IQR: $0.22 - 0.12s$; tilde denotes median), showing that the trajectories approximated a delayed pure pursuit course.”* We have also now added in the corresponding figure legend (Fig. 3) that *“The reasonably close linear association [between the delayed track angle and the line-of-sight angle] in (B) indicates that the attack trajectories of the hawks approximated a delayed pure pursuit course.”* Even so, the finding that $\gamma(t)$ is approximately proportional to $\lambda(t - \tau)$ is agnostic with respect to the guidance law that produces this. As we show elsewhere in the manuscript, a proportional pursuit (PP) guidance law (“turn at a rate proportional to the deviation angle between your target and your own velocity vector”) does about as good a job at modelling the trajectories as does the very proportional navigation (PN) guidance law (“turn at a rate proportional to the angular rate of the line-of-sight to your target in an inertial frame of reference”). A mixed PP+PN guidance law explains the data much better. There is no contradiction here: all three guidance laws can be used to produce a trajectory approximating a delayed pure pursuit; the mixed guidance law gives unequivocally the best fit to the data of the three.

To clarify these distinctions, we have added:

“Although this simple correlation analysis does not provide an explicit model of the dynamics, the fitted delay is nevertheless comparable to the 0.13s sensorimotor delay fitted in a steering controller used to model pigeons negotiating obstacles (12). ”

This is followed by the section on the different ways of implementing pure pursuit, which we now introduce with the sentence:

“The statement that the hawks' attack trajectories approximated a delayed pure pursuit course is agnostic with respect to the feedback system that implemented this.”

In addition, Fig. 3A,C show very similar histograms of the error for the pure pursuit and proportional navigation, although there is no similar data for the mixed model.

We have added the equivalent data for the mixed model to this figure, also marking the median prediction error and its bootstrapped 95% confidence interval.

Thus, there is no statistical support provided for preferring the more complicated mixed model over pure pursuit, or for distinguishing these three models. One might reasonably state these results as supporting the simplest model, which is pure pursuit, as in lines 128-129: “...hawks tune their mixed guidance law to produce a trajectory approaching a pure pursuit...”.

Thus, the conclusion should be restated as finding that all three models perform equally well at fitting the measured data. This study remains equally interesting, relevant and well-conducted if it supports the hawks studied using pure pursuit. There also may be a case that the mixed model is preferred as more biologically realistic, but the current manuscript does not make this argument. Possible additional data analysis (see below) might provide support for the mixed model or other alternatives. While this further analysis should not be required, I think it would strengthen the scientific argument.

In light of the various clarifications above, we hope the reviewer will now agree with our conclusion that the mixed PP+PN model fits the data significantly better than either PP or PN.

The authors could determine whether the mixed model is a better fit to more complex target trajectories by fitting each trajectory, as opposed to globally fitting all 50 at once to avoid over-fitting (line 97), using a measure of goodness-of-fit that accounts for the different number of fit degrees of freedom in each model. The analysis also could determine whether median fit error correlates with path complexity. For example, median fit error could be plotted and correlated with the ratio of distance traveled to start-to-end distance, number of turns, etc.

As described above, this is what we have now done. Accounting for the number of fitted parameters is difficult within the framework that we have used owing to the difficulty of specifying a likelihood function for the target of the optimization. However, we think that this is unnecessary given that the globally-fitted mixed PP+PN guidance model with only 3 fitted parameters fits the data just as well as the independently-fitted PP and PN guidance models with 100 fitted parameters each. Any reasonable method of penalising the likelihood function would arrive at the same qualitative conclusion as we have done, which is that the global PP+PN model is the better model.

2) On a related note, the manuscript states that parallel navigation has been ruled out, so the results of fitting the data using parallel navigation (and possibly a mixed parallel navigation and pure pursuit model) should be included. lines 128-129: "Why, though, should hawks tune their mixed guidance law to produce a trajectory approaching a pure pursuit, rather than a parallel navigation course?"

We agree that it was an oversight not to have shown directly that the birds did not fly a parallel navigation course. The difficult with testing this quantitatively is that whereas the general hypothesis of a pure pursuit course leads to an explicit prediction of how the track angle should vary through time, which is amenable to a correlational analysis (see above), the general hypothesis of a parallel navigation course leads only to the prediction that the track angle should remain constant. This is not a problem in practice, because plotting the lines-of-sight on a single trajectory chosen at random for each bird, as we have now done, makes it clear that the lines of sight are far from parallel, and hence that the birds are not using parallel navigation. We thank the reviewer for prompting us to include this new figure (Fig. 2).

Parallel navigation is not merely "geometric" (line 137): it is widely observed across many taxa and well-justified based on known mechanisms of visual processing and other sensory modalities (e.g. echolocation), as explained in many of the references cited. For example, a moving organism can implement a form of feedback-based parallel navigation by directly sensing optical flow of the target and making course adjustments to stabilize its visual angle.

Parallel navigation defined as a geometry in which the line-of-sight from pursuer to target remains parallel at all times. It is not itself a feedback law (in contrast to proportional navigation, which is the natural way of implementing parallel navigation), and the missile literature makes this distinction clearly. Nevertheless, we agree that several of the references that we cited at this point had implicitly or explicitly modelled a feedback law of some kind, and we have therefore modified the text to avoid implying otherwise by deleting the phrase "*These geometric descriptions are idealizations, however...*".

Assuming that parallel navigation was tested as a possible model but the results not reported, the authors could simply add the line-of-sight lines to their plotted trajectories or provide an equivalent measure, in keeping with all the refs cited that studied pursuit in animals (12-19).

See above. This is now done in a new figure (Fig. 2).

3) The manuscript implies that hawks were filmed chasing live "agile prey", but the hawks studied chased only an artificial target (a box filled with food):

Title: "Hawks steer attacks using a guidance system tuned for close pursuit of agile prey through clutter"

Amended to read: "Hawks steer attacks using a guidance system tuned for close pursuit of erratically maneuvering targets".

Abstract, lines 14-15: "Here we identify the guidance law that Harris' Hawks *Parabuteo unicinctus* use to chase jinking terrestrial prey in clutter." The manuscript should be revised to use a more accurate term such as "unpredictably-maneuvering target/dummy prey/falconry lure/etc.", as in refs. (13), (15).

Amended to read: "*Here we identify the guidance law of Harris' Hawks *Parabuteo unicinctus* chasing erratically maneuvering artificial targets, discussing how the structure and tuning of this guidance law relates to their typical hunting style of close pursuit, and considering how this compares to the guidance law used by falcons specialised on hunting in open environments.*"

Also, "jinking" is jargon unfamiliar to most readers of Nature and should be defined at first use.

We have deleted the term from the abstract, and defined it in the main text as follows: "the erratic maneuvers, or "jinks", of a terrestrial prey item".

These results are extremely interesting and relevant given this study design. There is much to be said for using a dummy target: it allows for reproducibility and replicability, varying trajectories on demand, and filming chases in a controlled location for imaging in three dimensions. Many other studies of animal navigation use them for this reason; this is especially suitable for organisms with poor visual acuity that may be unable to distinguish the difference from live prey, but this is not the case for birds.

However, because the target can influence the guidance law used (see below), the title, abstract and main text should describe the targets accurately, and note any related limitations when extrapolating the results to other contexts.

Title amended to read: "Hawks steer attacks using a guidance system tuned for close pursuit of erratically maneuvering targets".

The target of pursuit has been shown to influence the guidance law used in other systems. E.g., bats chasing prey use parallel navigation (ref (12)) to pursue live prey and pure pursuit to chase another bat (e.g., Chiu, Chen, et al. "Effects of competitive prey capture on flight behavior and sonar beam pattern in paired big brown bats, *Eptesicus fuscus*." *Journal of Experimental Biology* 213.19 (2010): 3348-3356.) Target motion should influence the choice of guidance law, as explained in the text.

We agree that target motion should influence the choice of guidance law, as explained in the text. However, the paper by Chiu *et al.* concerns an experimental setup in which pairs of bats competed to grab a stationary mealworm hanging from a thread. The objective of the inferred chasing behaviour of bat on bat is ambiguous in this setup, making the results of that study difficult to interpret in the context of individual prey pursuit and interception. Please see comments below for further discussion.

Target motion should influence the choice of guidance law, as explained in the text. However, it does not explain how closely the lure's motion resembles that of relevant prey. On video, the target looks like a tumbling box. Surely a visually-oriented hawk is aware it is not chasing a live animal.

It is true beyond reasonable doubt that a hawk could learn to discriminate between the dummy lure we used and a live target; whether the hawk is aware of that distinction is unknowable. This situation is certainly not unusual in a behavioral experiment. What matters is that the hawks were highly motivated to chase the targets: had they chosen to behave otherwise, they would have learned that they would be fed at the end of a day even if they did not bother to chase the targets at all.

How did the lure speeds, frequency and angles of maneuvering compare to those used by jackrabbits (the prey species of interest) during actual chases? The lure track diagram and videos make it seem

like the lure changed directly relatively infrequently and at a limited number of known locations compared to escaping rabbits; the hawks underwent training on the same course before filming (lines 204-205); and target velocity changes did not occur in response to the hawk's motion, unlike the case for live prey. Lines 204-205, 217-227 in the methods also mention path predictability as a possible confounder.

Our aim in this study was to model relevant kinds of target maneuver under controlled experimental conditions; not to simulate a jackrabbit. That said, we agree that it is useful to compare the dynamics of our target with those of a natural prey item, albeit that there appear to be none available for any species of jackrabbit. The only relevant measurements that we can find are of the European Hare *Lepus europaeus*, from the same genus. We have added the following sentences to the main text accordingly:

*“The target comprised a food lure, which was towed at speed around a series of pulleys to create a sequence of zigzagging turns that we randomised on each trial to prevent the hawks from learning the course (see Methods). The speed of the lure was adjusted continuously by the experimenter (median lure speed: 7.0ms^{-1} ; IQR: $9.7 - 4.1\text{ms}^{-1}$) to keep it ahead of the hawk until the moment of capture. The resulting motions were intended to mimic the erratic maneuvers, or “jinks”, of a typical terrestrial prey item (e.g. a jackrabbit, *Lepus* spp.). There is limited information available on prey performance, and we did not attempt to model the behavior of any particular prey item specifically, but the lure's turning performance was broadly comparable to that of the European Hare *Lepus europaeus*, which has been recorded making a 60° evasive turn on a 7m radius when fleeing a predator at 10ms^{-1} (9).”*

The course taken by the pulleys was made unpredictable by randomising the direction or occurrence of the switchbacks running down the middle of the course. Although this was explained in the Methods, the schematic illustrating the randomised experimental design had been consigned to Supplementary Information, and we have therefore moved this to the main text (as new Fig. 7) to clarify this important point, and have expanded the figure legend for clarity.

On a related note, earlier work not cited proposed a similar “integrated navigation law” in which the pursuer uses remembered landmarks to predict future motion using a combination of simpler guidance laws (e.g., Kong, Zhaodan, et al. "Perceptual modalities guiding bat flight in a native habitat." Scientific reports 6 (2016): 27252, Kong, Zhaodan, et al. "Optical flow sensing and the inverse perception problem for flying bats." Decision and Control (CDC), 2013 IEEE 52nd Annual Conference on. IEEE, 2013; Kong, Zhaodan, et al. "Perception and steering control in paired bat flight." IFAC Proceedings Volumes 47.3 (2014): 5276-5282.)

As we explain in the text, whilst it is possible that the hawks could have learned the rule that the target would always move toward the middle of the course from the pulleys at the periphery, the actual course taken by the pulleys was made unpredictable by randomising the direction or occurrence of the switchbacks running down the middle of the course. The combination of randomised pulley assignment, randomised pulley direction, randomised lure start-positions, and naturally variable hawk start-positions was such that the bird would never have flown the same course more than once. These earlier works are great papers, but we do not think they are directly relevant to our study, which was specifically designed to produce reactive responses.

4) Although the title and abstract indicate that the hawks pursued target through clutter, the manuscript does not describe anything approximating clutter (e.g., brush, tall grass, branches, etc.) as being present during chases. Instead, the methods section (lines 215-217) describe the lure as moving across a “mown grassy field” guided by a series of winches and pulleys. This clutter-free environment also is depicted in the videos and images shown. Either the paper should explain how clutter was included in the experimental design or else refrain from stating that chases in clutter were studied. Either way, it makes sense for the authors to discuss which navigation laws might work best in clutter in the context of their findings.

This is an important point, and we had not intended to imply that we had modelled clutter: the only actual clutter present in the test environment was the pulleys and the tunnels containing them. The point was rather that Harris' Hawks are adapted to flight in cluttered environments, and that the mixed guidance law that we identified in an uncluttered environment is much better suited to flight through clutter than the pure proportional navigation guidance law that we have identified previously in peregrines. This requires more explanation than we have space for in the title, which we have therefore amended to read:

"Hawks steer attacks using a guidance system tuned for close pursuit of erratically maneuvering targets".

Likewise, the revised abstract now reads:

"Hawks use a mixed guidance law coupling low-gain proportional navigation with a low-gain proportional pursuit element. This guidance law promotes tail-chasing and is not thrown off by erratic maneuvers, making it well suited to the hawk's natural hunting style, involving close pursuit of agile prey through clutter."

and we have now prefaced the relevant paragraph of the Discussion with the sentence:

"We did not directly study the role of habitat clutter in this experiment, but the clutter that is typical of the habitats in which hawks hunt offers another functional account of why a mixed guidance law might be advantageous."

5) Ref. (6) proposed and provided evidence that hawks pursuing live maneuvering prey through cluttered natural environments use a mixed strategy (initial parallel navigation that transitions into pure pursuit at close quarters) during a large fraction of pursuits, and explicitly noted that proportional navigation might provide a more general model. This is so similar to the hypothesis tested in this study that it ought to be mentioned in the introduction. Instead, the introduction suggests that this research supported pure pursuit only:

lines 32-34: "Hawks may be expected to use a different guidance law, because their attacks often follow a pure pursuit course, in which flight is aimed directly at the target (6)."

and a more accurate summary is left for the discussion:

lines 135-136: "Previous research on another hawk species found a mix of both (6), which might be explained by the use of a mixed guidance law combining elements of PP and PN"

We did not mean to imply otherwise, and have clarified the introductory sentence along the same lines as we had already done in the discussion. The sentences in the Introduction now read:

"At high enough gain, this feedback system produces an attack trajectory called a parallel navigation course, in which the geographic direction of the line-of-sight remains approximately constant, causing the attacker to intercept its target by heading it off. This is in contrast to the simpler geometry of a pure pursuit, in which flight is aimed directly at the target, causing the attacker to intercept its target in a tail chase. Hawks may be expected to use a different guidance law to Peregrine Falcons, because their attacks have been found to involve a mix of different guidance behaviors, combining elements of parallel navigation with elements of pure pursuit (8)."

The revised version should credit this earlier work and other studies that use similar integrated/mixed navigation models, including the references cited at the end of comment 3 above and ref (18).

For the reasons explained above, we do not think that the references cited in comment 3 are directly relevant to our study. In particular, the "mixed" aspect of the guidance in Kong et al's study of bats emerging from their roost relates to the particular combination of reactive obstacle avoidance and the learning of stereotyped flight trajectories night after night. This is a very different thing to what we have studied and shown, which is the combination of two distinct feedback elements in a single mixed guidance law. We therefore feel that to add these citations as requested would confuse our manuscript.

Minor revisions:

1) Harris' Hawks are pack hunters in their native environments. They presumably have evolved to solve a very different problem from other raptors, all of which hunt solo in most cases. How might this unique cooperative hunting strategy influence the navigation law they use for chasing prey? (Note that this point can be phrased to enhance the broader applicability of their findings.)

This is an interesting question, which we would love to study, but it is not one that we think we can tackle with the present dataset. Cooperative hunting is far from ubiquitous in Harris' Hawks, having only been identified in certain sub-populations whose challenging desert environment presumably favours such behaviour. Where it is observed, the behaviour is a tactical one in which, for example, one hawk may wait perched ready to attack whilst another flushes the prey from cover. We do not consider it obvious what form of guidance would be optimal in this circumstance, or how this might differ from the forms of guidance favoured in solitary hunting behaviours. Finally, whilst Harris' Hawks were the first raptors in which cooperative hunting was identified, and are still widely mentioned as the only species to display such behaviour, cooperative hunting has since been reported in several species of falcon. This interesting feature of their behaviour is therefore no longer considered unique.

This also merits a comparison with the results of studies of bats chasing prey cited above that found bats use parallel navigation to chase prey but pure pursuit to chase another bat during cooperative hunting. (See ref. (12) and Chiu, Chen, et al. cited above). Given that the Harris' Hawks use of pure pursuit, might this not arise from their similar use of cooperative hunting?

The study by Chiu et al. studied competitive, as opposed to cooperative, interactions between bats. It is therefore far from clear why – or perhaps, whether – either bat was actually chasing the other, given that the experiment had been deliberately set up so that they would compete to capture a stationary mealworm with which they were presented simultaneously. We do not think it will shed much light to have to tackle these ambiguities in the present manuscript.

2) The methods section explains that the trajectories analyzed were in the ground plane (an appropriate choice here), but this is worth mentioning briefly in the main text or main figure captions.

We have added the following sentence to the main text: *“Because the recorded attack trajectories were always close to planar, we projected them onto the two-dimensional ground plane defined by the lure’s trajectory prior to further analysis (see Methods).”*

3) The following sentences imply that the hawk’s gaze-direction was measured:
line 48-50: “The hawks took off as soon as the lure began moving, flap-gliding to close range, and banking to turn whilst keeping their eyes level and their gaze directed at the target”
line 118: “under the assumption that the head tracks level and tracks the target closely”

The authors do not need to measure gaze-direction for this study. It seems unlikely gaze-direction can be determined from the videos, given the low resolution of the hawk’s head and eyes, its varying visibility and the known complexity of hawk visual systems; head nystagmus is actually violated at the end of one video, although it’s well established in the literature during level flight. The gaze-direction is not the same as the angle between the positions of the hawk’s body and the lure (conventionally referred to as “line-of-site” in the literature; see line 281 in the methods).

These lines should be rephrase to avoid implying it was measured.

We have edited the corresponding figure legend (Fig. 1) to indicate that the head is held with the eyes approximately level, as is visible from the video frames at full zoom. We have edited the first of the aforementioned sentences to read *“...whilst appearing to keep their eyes level and their gaze directed at the target”*. The second of the sentences now reads: *“This angle could be estimated in various ways, but will be similar to the angle between the body axis and the sagittal plane of the head if the*

head is assumed to be kept level and to track the target closely." where we have deleted a reference to a figure to avoid implying that the qualitative evidence that this provides was sufficient to justify the conditional statement that we made.

Suzanne Amador Kane

We are grateful to you for your careful consideration of our manuscript, for your time and effort, and also for your candour. We hope you will agree that our manuscript has been enhanced by the changes that we have made in light of your comments, and that it is now in a form that you can recommend for publication in *Nature Communications*.

Reviewer #3 (Remarks to the Author):

The paper claims that Harris' hawk uses a combined proportional and pursuit based guidance strategy to intercept fleeing prey. The outcomes are convincing and I have made some suggestions below about clarifying some aspects. Appropriate references are made to the existing literature. This research is difficult to undertake and I believe the very best effort possible and practical has been made here, so I would not recommend any new or different experimental work. The statistical analysis is convincing and the results are quite clear and method reproducible by those with access to the same resources.

We are grateful to the reviewer for their thorough review and positive assessment of our manuscript, and for their detailed and helpful comments. We hope they will agree that we have addressed these all through our revisions to the manuscript, and our point-by-point response below.

There is an early discussion of evolution of different raptor species, leading the reader to expect a strong hypothesis test related to evolution which does not happen. It is not wrong, but perhaps either more or less needs to be said.

We have opted to say less, deleting the phrases "*and may perhaps have been an evolutionary precursor to the other aerial targeting behaviors such as perching that are characteristic of landbirds. Here we ask whether the guidance systems of hawks and falcons have diverged together with their respective hunting styles.*" We have moved the amended sentence "*The higher-level clades containing hawks and falcons diverged >60mya, but as their common ancestor with other landbirds is also thought to have been an apex predator (7, 8), it is reasonable to suppose that their raptorial guidance systems might share a common evolutionary origin.*" to an earlier point in the Introduction, so that it is clearer that this simply represents relevant background information. Finally, we have added the new summary sentence: "*Here we identify the guidance law of Harris' Hawks *Parabuteo unicinctus* chasing erratically maneuvering artificial targets, discussing how the structure and tuning of this guidance law relates to their typical hunting style of close pursuit, and considering how this compares to the guidance law used by falcons specialised on hunting in open environments.*" to make the scope of the comparison clear.

In figure 1 The choice of lambda and gamma for key variables is unfortunate since it leads to visual confusion for many readers.

We do see the point, but had deliberately used this notation for consistency with our previous work on peregrines, and for consistency with a well-known text in missile engineering on which the theory of this earlier paper was based. We would therefore be reluctant to change the notation in this paper.

Starting line 53 Is the 0.16s sensorimotor delay based on a reasonable foundation? non-neuroscientists may be concerned about what could be seen as a free variable (I think it is a reasonable number for the several synapses and neuromuscular junction involved).

The delay term is effectively a black box covering both the sensorimotor system itself and the aerodynamics that transfer muscle action to external force, and it is important to note that the fitting of the delay at this point in the manuscript is in relation to a correlation between variables rather than in relation to the identification of a dynamical system. It is therefore difficult to answer this question from first principles, but to alleviate any concerns we have added: *“Although this simple correlation analysis does not provide an explicit model of the dynamics, the fitted delay is nevertheless comparable to the 0.13s sensorimotor delay fitted in a steering controller used to model pigeons negotiating obstacles (12).”*

Line 66-68 it might be best to not say this regarding the missile guidance field because it does not accurately characterize what they are actually trying to achieve and the characteristics of their targets and where their interests lie. I don't think it would take anything away from the manuscript if it was not there.

We were probably trying to say too much in one short sentence here, but think it appropriate to cite the few studies that have used this particular form of mixed guidance law previously, so have simplified the sentence to read: *“Another possibility is to use a mixed guidance law $\dot{\gamma}(t) = N\dot{\lambda}(t - \tau) - K\delta(t - \tau)$ combining PP and PN (5), which has already been tested in several other contexts in the missile literature (9-11).”*

Figure 3 has very flat error response with regard to time delay for all error surfaces,

This is an important point, which we have noted explicitly by adding: *“Although the optima for the guidance constants N and K were quite sharply defined, the optimization surface was rather flat with respect to the delay τ , so we have lower confidence in stating a precise value for the latter.”*

I notice in lines 90 to 91 a comment that the PP response was effectively instantaneous, which is not possible without some sort of prediction or tracking which in itself would then be part of the guidance law. This seems to be a complex situation with a simplistic statement to explain it.

We have added to these sentences as follows: *“Interestingly, the majority of the PP simulations entailed an effectively instantaneous response ($\bar{\tau} = 0.00s$; IQR: 0.09 – 0.00s; Fig. 3C), which in practice would imply the presence of a predictive element to overcome the inevitable sensorimotor delay. In contrast, the PN simulations typically involved a more delayed response, with a median fitted delay of $\bar{\tau} = 0.10s$ (IQR: 0.19 – 0.00s; Fig. 3F), which is of similar magnitude to the sensorimotor delay identified in previous studies of avian guidance behaviors (12).”*

At line 98 I don't think the word “Remarkably” adds any value.

Deleted.

102 the recognition that there is individual variability will immediately have readers wondering if the entire behaviour is learned from recent experience and the nature of the problem, or if the behaviour is hard-wired neural circuitry. A sophisticated vertebrate predator might have learned or at least tuned their innate behaviour to their environment.

We agree, and apologise that we had not provided the statistics needed to assess this in our original submission. We have now added:

“The mean ranks of the independently-fitted values of N varied significantly between individuals (Kruskal-Wallis test: $\chi^2(4) = 12.96$, $p = 0.01$), but a post hoc test only found evidence of one significant pairwise difference between birds (Tukey's HSD at $\alpha = 0.05$), so we do not attribute much importance to this result. There was no evidence of any significant variation in the independently-fitted values of K between individuals (Kruskal-Wallis test: $\chi^2(4) = 3.16$, $p = 0.53$).”

at the relevant point in the Results. We do not think that these results are sufficient to warrant the drawing of any very strong conclusions regarding inter-individual variability, particularly in light of how closely the flights are modelled by the unique globally-fitted mixed guidance law.

Line 119 the vestibular system might be the source of the inertial reference frame, but there are visual possibilities here as well that seem more likely in a visual pursuit, or a fusion of both

We agree, and have added to this sentence so that it now reads:

“Under the same set of assumptions on head tracking, the line-of-sight rate could be estimated either by integrating the angular accelerations sensed by the vestibular system, or by making direct use of the rotational optic flow cues produced by the head’s self-motion relative to a fixed visual background.”

For the discussion in the area of 130 to 140, it is worth pointing out the obvious that some of the prey species of vertebrate predators may be capable of sophisticated counter measures, a situation that is not tested in this work, since I assume the jinking of the target is open loop with regard to the hawk trajectory.

This is an important point, which we are studying extensively in separate computational work using a deep learning framework to optimise the performance of predator and prey. There is much to say on this elsewhere, but for now we have simply added the following caveat to the discussion: *“It is an open question how these different kinds of attack behaviors might perform in response to closed-loop evasive maneuvers”*.

The conclusion area from 164 to 184 seems reasonable. I do wonder about headlining the drone issue as the last sentence. For a lot of reasons drones are not a difficult target once detected and tracked, and are not likely to be for some time.

We have deleted the sentences referring to drones from the Abstract and Discussion.

I believe this could be a classic work about animal pursuit. But I think there are stylistic elements that need to be worked on. The paper appears to be seeking justification for itself in other fields (missile guidance and counter-drone applications). This actually detracts from it being a classic work, since it might be criticized by those in other fields for aspects that are not really part of the hypothesis.

We are grateful to the reviewer for their generous assessment of our manuscript. We have removed all reference to drones accordingly, and now refer to the missile literature only where it is appropriate to do so in giving credit for previous work where it is due.

REVIEWERS' COMMENTS:

Reviewer #1 (Remarks to the Author):

The authors have fully met all of the concerns I expressed in my original review, as well as those of the other reviewer. They have greatly strengthened their original conclusions in doing so, clarified their arguments in ways that will speak to a wide readership, and added to the already considerable interest and significance of the research. They have quite reasonably opted not to include some additional references and material I suggested, for sound reasons in each case. I therefore most strongly recommend that you accept the revised manuscript without further revision.

Reviewer #2 (Remarks to the Author):

I am happy with the changes that have been made. All of my original concerns have been comprehensively answered and substantial clarifications have been made. I believe this manuscript is now acceptable for publication in Nature Communications.

Javaan Singh Chahl